# Unconscious reinforcement learning of hidden brain states supported by confidence

Aurelio Cortese 1✉, Hakwan Lau 2,3,4,5 & Mitsuo Kawato 1,6✉

Can humans be trained to make strategic use of latent representations in their own brains? We investigate how human subjects can derive reward-maximizing choices from intrinsic high-dimensional information represented stochastically in neural activity. Reward contingencies are defined in real-time by fMRI multivoxel patterns; optimal action policies thereby depend on multidimensional brain activity taking place below the threshold of consciousness, by design. We find that subjects can solve the task within two hundred trials and errors, as their reinforcement learning processes interact with metacognitive functions (quantified as the meaningfulness of their decision confidence). Computational modelling and multivariate analyses identify a frontostriatal neural mechanism by which the brain may untangle the 'curse of dimensionality': synchronization of confidence representations in prefrontal cortex with reward prediction errors in basal ganglia support exploration of latent task representations. These results may provide an alternative starting point for future investigations into unconscious learning and functions of metacognition.

[1] Computational Neuroscience Laboratories, ATR Institute International, 2-2-2 Hikaridai, Seika-cho, Soraku-gun, Kyoto 619-0288, Japan. [2] Department of Psychology, UCLA, 1285 Franz Hall, Los Angeles, CA 90095, USA. [3] Brain Research Institute, UCLA, 695 Charles E Young Dr S, Los Angeles, CA 90095, USA. [4] Department of Psychology, University of Hong Kong, 627, The Jockey Club Tower, Pok Fu Lam Rd, Pok Fu Lam, Hong Kong. [5] State Key Laboratory for Brain and Cognitive Sciences, University of Hong Kong, 5 Sassoon Rd, Pok Fu Lam, Hong Kong. [6] RIKEN Center for Advanced Intelligence Project, ATR Institute International, 2-2-2 Hikaridai, Seika-cho, Soraku-Gun, Kyoto 619-0288, Japan. ✉email: cortese.aurelio@gmail.com; kawato@atr.jp

We consciously perceive our reality, yet much of ongoing brain activity is unconscious[1,2]. While such activity may contribute to behaviour, presumably it does so automatically and is not utilized with explicit verbal strategy. Can humans be trained to make rational use of this rich, intrinsic brain activity? From the outset, this problem is challenging because the relevant activity is often high dimensional. Given so many latent dimensions, how can subjects know what to learn? This question is more general than it may appear: having to probe through vast search spaces, among many possible states for efficient learning, is widely recognized as one of the core challenges in reinforcement learning (RL), or the 'curse of dimensionality'[3,4]. Particularly so in the brain, where sensorimotor learning is a classic example[5], but likewise most social and economic decisions are difficult in the sense that there is no external and explicit state which is relevant to RL.

Previous studies have shown that RL can operate on external masked stimuli[6–8]. In those studies, the relevant subliminal information was driven by a simple visual stimulus, which carried only a single bit of information. Other studies have shown that human participants can decide advantageously before consciously knowing the strategy[9]. Here we address a more challenging question with a technique based on internal multivariate representations. Specifically, subjects have to learn a hidden state (the internal multivariate representation) with many dimensions generated stochastically within the brain.

We draw inspiration from brain–computer interface (BCI) studies in monkeys[10,11]. Using a decoder (a machine learning classifier), subjects' neural activity patterns (in either the prefrontal cortex—PFC, or visual cortex—VC) measured with functional magnetic resonance imaging (fMRI) determine in real-time the 'state' of an RL task (Fig. 1, Methods). The decoder is based on representations of right and left motion direction, so as to have a clearly separable boundary also at the neural level. To construct the decoder, we used fMRI data collected a week before the main RL task (Supplementary Fig. 1, Methods). VC is chosen because it is the first stage of cortical processing for visual information, and its features are known to be mainly linked to simple, objective aspects of stimuli[12,13]. PFC representations on the other hand are thought to be mainly related to subjective aspects of the perceived stimuli[14,15]. Based on these functional differences we predict different learning results depending on where the decoder was built.

Each trial starts with a blank interval, followed by random dot motion (RDM) with 0% coherence displayed for 8 s. After stimulus presentation subjects report what they perceive as rightward or leftward motion (discrimination), rate their confidence in their choice and lastly, gamble on two options (A or B) that can potentially lead to reward (30¥/0.25$). Unbeknownst to the subjects, whether it is option A or B that is more likely to be rewarded (i.e. the optimal action) is determined by a multidimensional pattern of their own brain activity measured at pre-stimulus time. That is, these patterns are input to the decoder, which categorizes them into latent RL states. Importantly, the purpose of the decoders here is not to find motion direction information in brain activation patterns. Rather, the purpose of the decoders is to divide brain activity into two classes, so as to define the latent RL state unconsciously. Because the time of decoding is pre-stimulus and the ensuing stimulus itself carries no direction information, the decoder alone defines the latent state from stochastic brain activity, along a predetermined classification boundary. Such multidimensional patterns are known to represent information that is generally below consciousness[1,16–19].

Although not fully unconstrained, spontaneous activity of neural populations is less structured than activity generated by specific sensory inputs[20,21]. The setting adopted here implies that the search for optimal policies in RL should explore a hidden, relevant state among a relatively high number of possible states defined by patterns of neural activity. Even the best artificial intelligence algorithms struggle to handle such problems in everyday, real-world problems when the training sample is small[22].

Given the unconscious nature and the high dimensionality of the neural activity used as task contingencies, it may thus seem improbable that subjects can learn to perform advantageously. Besides, previously we have proposed that solving such problems may correlate with the mechanism of metacognition, manifested as confidence judgements, and illustrating the ability of an agent to introspect and track its own performance or beliefs[23–25]. Recurrent loops linking frontal and striatal brain regions could support this interaction between RL and metacognition[24,26,27]. Although seemingly counterintuitive, metacognition can exist in the absence of awareness, as unconscious metacognitive insight: human subjects can track their own task performance while claiming to be unaware of the stimuli or the underlying rule[9,28–30].

The main objective of this study is to test if humans can learn a task in which the information that determines the RL states is (a) high dimensional and (b) unconscious. As a corollary, we ask whether metacognition is involved in such a learning scenario.

To anticipate, we find that subjects can learn the gambling task. Moreover, rather than a simple learning effect in selecting the optimal action, we uncover that subjects' metacognition (quantified as their confidence in their choices) correlates with RL processes, both at the behavioural and neural level. Surprisingly, there are no differences between the two groups of subjects—decoder in VC vs. PFC—in terms of learning performance, indicating that the mechanism may be general enough to support learning in any brain region where neural activity is, or becomes, relevant to earn rewards.

## Results

**Behavioural accounts of learning.** We first evaluated whether human subjects displayed any evidence of learning the reward-maximizing action-selection task over the course of about two hundred trials. To do so, empirical optimal action-selection rates were compared to a chance level of 0.5, the rate attained if actions were randomly selected at every trial. For all tests against chance, we utilized full linear models, with the intercept as difference from chance and subjects as random effects. In session 1 the rate was not different from the random model (Fig. 2a, $\alpha = 0.024$, $t_{17} = 1.54$, $P_{(FDR)} = 0.14$). Subjects selected their actions significantly better than chance in session 2 (Fig. 2a, $\alpha = 0.039$, $t_{17} = 3.62$, $P_{(FDR)} = 0.003$). The increase from the first to the second session was a trend and not significant (Fig. 2a, one-tailed sign test, sign = 6, $P_{(unc.)} = 0.12$), but averaging the rates over the first two sessions confirmed overall above-chance performance (Supplementary Fig. 2a, $\alpha = 0.032$, $t_{17} = 3.17$, $P_{(unc.)} = 0.006$). This happened despite the fact that decoded state information was not physically presented to the subjects, and that their discrimination performance was lower and indistinguishable from chance (Supplementary Fig. 2b). We confirmed that discrimination performance was indeed different from optimal action-selection performance in the first two sessions (linear mixed effects [LME] model, fixed effects 'task' and 'session'; significant effect 'task' $\beta = -0.026$, $t_{69} = -2.24$, $P = 0.028$, Supplementary Table 1). A regression analysis of $p$(opt action) vs. $p$(corr discrimination) in sessions 1–2 resulted in a trend that better discrimination was associated with better gambling choices. But more importantly, the intercept was significantly larger than 0: optimal action-selection rate had a higher baseline than correct discrimination (Supplementary Fig. 2d, linear regression, $\alpha = 0.026$, $P = 0.0077$,

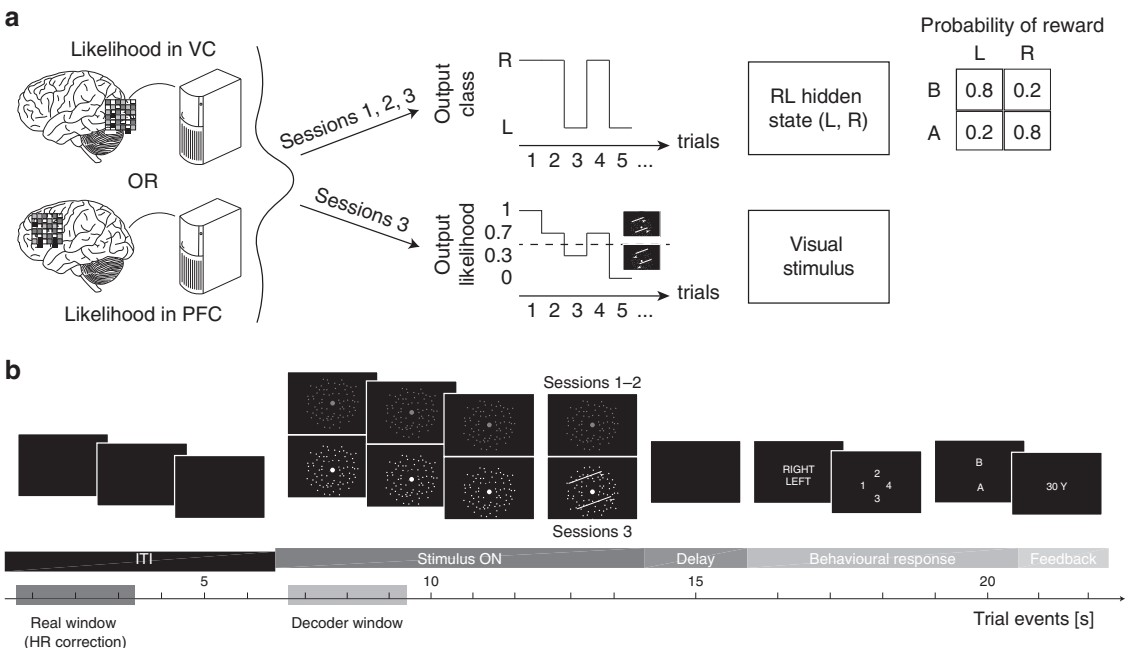

**Fig. 1 Design of the hidden-brain-state reinforcement learning task.** Subjects ($N = 18$) were assigned to one of two groups, which differed in the brain region targeted by their decoder: visual cortex (VC, $N = 9$) or prefrontal cortex (PFC, $N = 9$). For all analyses, the brain region was treated as a between-subjects factor; unless this factor displayed a significant effect, subjects were pooled into one cohort. **a** The learning task consisted of three consecutive sessions. In each session, decoding was performed with fMRI multivoxel patterns; the decoder output was used in real time to determine the RL state on a trial-by-trial basis. In a given RL state, only one action was optimal, with a high probability (0.8) of reward, while the other action had low reward probability (0.2). In the last (control), third session the output likelihood was also used to proportionally define the motion direction of the visual stimulus. Even in the last session, early trials had very low coherence, and only the latter half of the session had trials with coherence high enough to be easily detected and for subjects to consciously learn the rule. **b** Each trial started with a blank intertrial interval (ITI, 6 s). Random dot motion was then shown for 8 s (Stimulus ON). On the first two sessions, the motion was entirely random and the dots were dim (20% of maximum), while on the third session the last 2 s had increasingly higher coherence (partially determined by the decoder's likelihood). Subjects then had to report the direction of motion (the latent state), their confidence in their choice, followed by a gamble on one of two actions (A or B). After action selection, the outcome for the current trial (reward: 30¥/0.25$, or no reward: 0¥/$) was shown on the screen. Accounting for the haemodynamic delay meant that decoding was performed on data corresponding to the ITI. This ensured that mental imagery or illusory perception could not index the latent state determined by the decoder from neural activity. HR haemodynamic response delay, L left, R right.

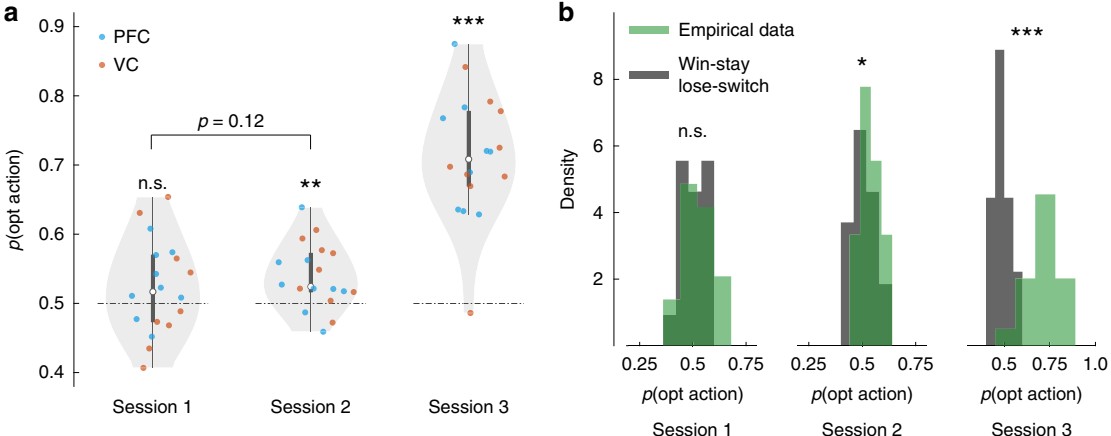

**Fig. 2 Learning to choose optimal actions. a** Subjects' probabilities of selecting the optimal action (the one that was more likely to be rewarded on a particular trial) in each session. The shaded areas in the violin plots represent the population spread and variance, the white dot the median, the thicker line the interquartile range, and coloured dots individual subjects. Within-session statistical test against chance: full linear model, with the intercept as difference from chance (two-sided $p$ values, FDR corrected). Between-session statistical test of difference (sessions 1 vs. 2): sign test (one-sided $p$ value, uncorrected). **b** For each subject, a control $p$(opt action) was computed according to a win-stay lose-switch heuristic. Under this strategy, an agent repeats the same action if it was rewarded in the previous trial, and switches otherwise. Grey histograms represent the probability density function (PDF) of $p$(opt action) from the win-stay lose-switch strategy, while coloured histograms represent the PDF of actual subjects' $p$(opt action) rates. Within-session statistical test of difference: sign test (two-sided $p$ values, FDR corrected). The experiment was conducted once ($n = 18$ biologically independent samples). **a** [n.s.]$P = 0.15$, **$P = 0.003$, ***$P = 2.9 \times 10^{-8}$; **b** [n.s.]$P = 0.48$, *$P = 0.019$, ***$P = 0.0004$.

slope $= 0.35$, $P = 0.089$), consolidating the LME results reported above. This dissociation between discrimination and action selection was likely due to the absence of a direct reward for discrimination choices.

A third session in which visual stimuli explicitly carried motion direction information inferred from brain activity by the decoder was also included as a control. In this session, the motion coherence slowly increased from 0% to higher values over trials within the first half of the session, remaining high henceforth (Fig. 1, Methods). The correct state was easily discriminated (Supplementary Fig. 2b), and most subjects consciously discovered and reported the action-selection rule; e.g., state$_{left} \rightarrow$ action B, state$_{right} \rightarrow$ action A (16 out of 18, binomial test against 0.5, $P = 0.001$), resulting in consistently high selection rates for the optimal action (Fig. 2a, $\alpha = 0.212$, $t_{17} = 10.32$, $P_{(FDR)} = 2.9 \times 10^{-8}$).

Nonetheless, other behavioural strategies could account for the action-selection performances in sessions 1 and 2 (besides those described hereafter, additional points are addressed in Supplementary Note 1). We first tested one simple, alternative model: the win-stay lose-switch heuristic. This strategy determines the action at trial $t$ as depending on the outcome at $t-1$: repeat the same action if reward was obtained, switch to the alternative action otherwise. Win-stay lose-switch performance was computed with subjects' session data (actions, rewards, and states); the starting point was the first action taken by the subject. Figure 2b indicates that in session 1 subjects' behaviour could be explained by this model (two-sided sign test, sign $= 7$, $P_{(FDR)} = 0.48$), but that in session 2 the performance attained was significantly lower than the real performance (two-sided sign test, sign $= 3$, $P_{(FDR)} = 0.019$). Session 3 confirmed the result anticipated in session 2 (two-sided sign test, sign $= 1$, $P_{(FDR)} = 0.0004$).

One further possibility could be asymmetric learning of a single latent state, which could then be repeatedly generated by the brain (e.g., that state 'Right' was paired with action 'A'). This can be easily tested: in the presence of asymmetric learning, one should not only see the emergence of a latent state bias, but also a steady increase in state bias over time. Because the bias can be directed towards either one of two states, we define here state bias as the unsigned difference between the number of occurrences of each state, normalized by the number of trials. Figure 3a illustrates two state occurrences time-courses in each session (example subjects S2 and S10), while Fig. 3b, c display the state-bias traces and sessions' means for all subjects. Surprisingly, state bias was non-zero from the beginning, but also constant in time, invalidating the hypothesis that the brain simply learned an asymmetric association and induced one state over and over. Rather, the state bias was an inherent feature of the latent state estimation through decoding.

**Computational accounts of learning**. The implication of these results is that any early above-chance action-selection performance likely depended on RL operating unconsciously. Nevertheless, RL itself could have resulted from two non-exclusive processes: (1) a noisy state-dependent RL process (RL$_{sd}$) where the update rule depends on both estimated latent states (defined as the decoder output) and actions; (2) a state-free RL process (RL$_{sf}$) where the agent simply selects the action associated with the highest expected value (regardless of the latent state). The RL$_{sd}$ model assumes that the agent performs some noisy inference/estimation of the latent brain activity. The RL$_{sf}$ model, conversely, is a naive process, in which the agent merely considers its actions' outcome. We therefore utilized computational modelling based on the noisy RL$_{sd}$ and RL$_{sf}$ variants of the standard Q-RL algorithm[4] (Methods, Eqs. 2–3). The two learning models were fitted to subjects' behavioural data, and free parameters were

estimated by minimizing the negative log-likelihood. To note, the noisy RL$_{sd}$ was designed such that, on a subset of trials determined by the amount of noise, the update was not based on the real RL latent state, but on the alternative state. The noise level was estimated and averaged over 100 resampling runs (see 'Methods').

Before formally comparing the two learning models, we can test a small, but important, prediction that arises from the main difference between the models, i.e., whether the latent state is considered or not in the model. In the presence of state bias (as established earlier, Fig. 3b, c), an agent using RL$_{sd}$ would be unaffected—because actions are contingent to the states themselves; conversely, an agent following a pure RL$_{sf}$ strategy would learn to choose the action associated with the biased state most of the time. Therefore, we expect the strength of the latent RL state bias to predict action-selection performance, if the RL$_{sf}$ is the main mechanism behind the learning behaviour. Averaging data from both session 1 and 2 argue against such interpretation (Fig. 3d). Yet, for most subjects (17/18) the bias was constant in sign in the first two sessions (Supplementary Fig. 3a), raising the possibility that partial learning of latent state bias from session 1 could have transferred to session 2. Replotting Fig. 3d session-by-session resulted in a non-significant reversal of the sign of the correlation (Supplementary Fig. 3b, two-sided $z$-test statistics on Fisher-transformed $r$, $z = -1.3$, $P = 0.19$). So far, these results are unfavourable to a state-free learning strategy.

The modelling approach allowed us to directly compare the two RL strategies. A simple visual inspection (Fig. 3e, example subjects S2–S10) suggests that action selection time-courses from the RL$_{sf}$ model (black lines, top) appear qualitatively different from the subjects' own time-courses (blue lines), while those from the noisy RL$_{sd}$ look more similar (grey lines, bottom). Akaike Information Criterion (AIC)[31] was computed for each model, subject and session. In all sessions, the noisy RL$_{sd}$ had lower total AIC (Fig. 3f left, $\Delta$AIC $< 0$: $\Sigma$AIC noisy RL$_{sd} < \Sigma$AIC RL$_{sf}$, session 1 $\Delta$AIC $= -23.6$, session 2: $\Delta$AIC $= -35.6$, session 3: $-563.5$). We also considered AIC at the subject level to obtain a more nuanced picture (Fig. 3f, right). AIC for noisy RL$_{sd}$ was similar to AIC for RL$_{sf}$ in session 1, significantly lower in session 2, but also significantly lower when taking the average of sessions 1–2 (full linear models: session 1, 9/18 AIC$_{sd} <$ AIC$_{sf}$, $\alpha = -1.31$, $t_{17} = -1.66$, $P_{(FDR)} = 0.12$; session 2, 14/18 AIC$_{sd} <$ AIC$_{sf}$, $\alpha = -1.98$, $t_{17} = -3.16$, $P_{(FDR)} = 0.009$; mean session 1–2, 15/18 ACI$_{sd} <$ AIC$_{sf}$, $\alpha = -1.65$, $t_{17} = -3.93$, $P_{(unc.)} = 0.001$, session 3, 17/18 AIC$_{sd} <$ AIC$_{sf}$, $\alpha = -31.30$, $t_{17} = -4.30$, $P_{(FDR)} = 0.001$). The same results were obtained when AIC was computed using the normalized log-likelihood (Supplementary Fig. 4a). Finally, in accordance with our intuition, the estimated noise level in RL$_{sd}$ was lower in session 3 compared with sessions 1–2 (Supplementary Fig. 4b). These results indicate that exploration of latent RL states within high-dimensional brain dynamics did occur (to some extent) even during the first two sessions of the gambling task.

**Perceptual confidence correlates with RL**. Since subjects do not have access to the decoder boundary, this is a computationally complex problem. The brain essentially has to find a latent low-dimensional manifold among high-dimensional subconscious brain dynamics only by trial and error. How can this curse of dimensionality be resolved?

The conceptual model introduced earlier[24] speculates that metacognition may be involved in this process. Although the design utilized here cannot afford to disambiguate the direction of the arrow of causality between metacognition and RL processes, we can, at a minimum, investigate whether the two become correlated during learning. It may sound odd that while

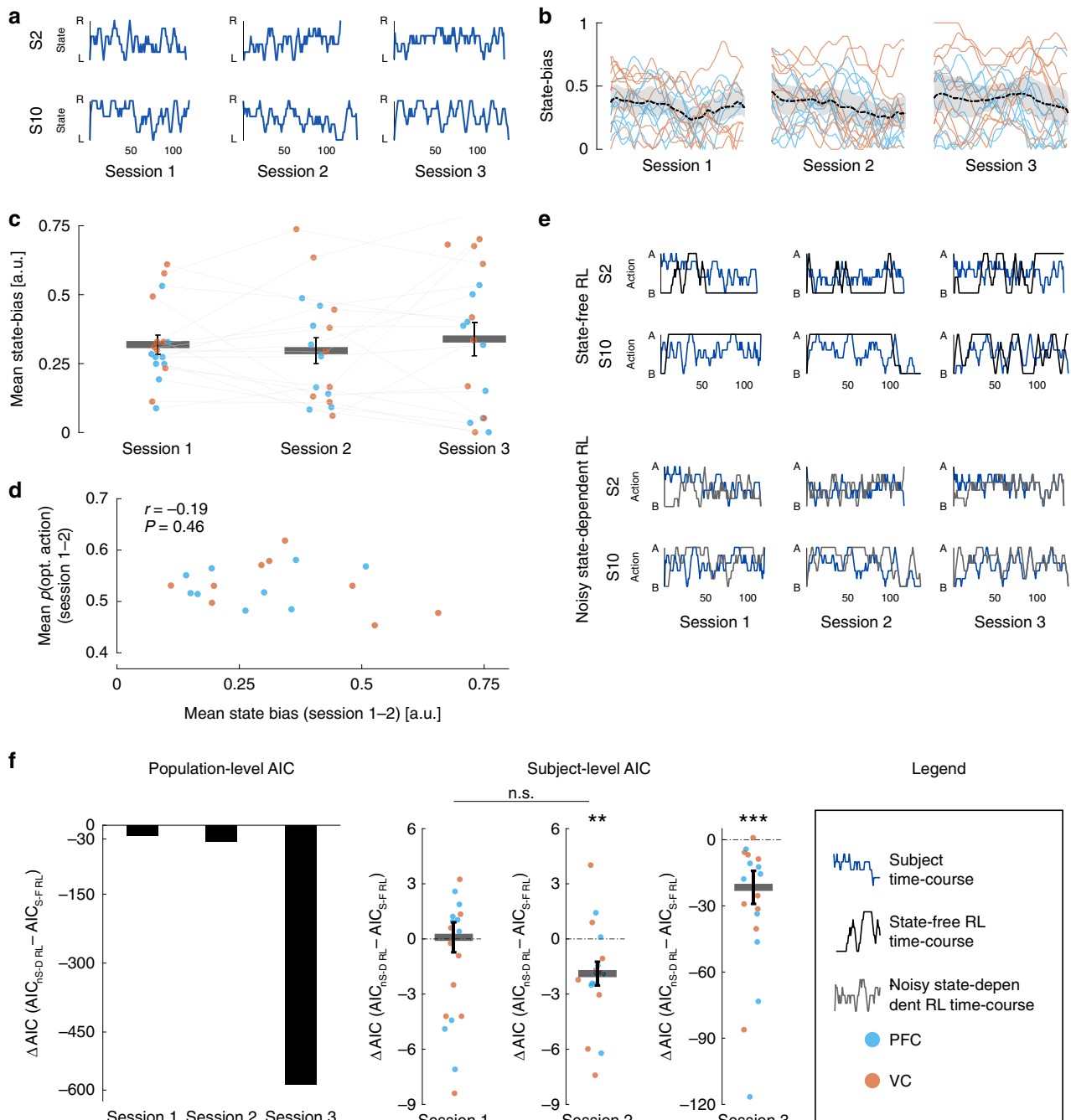

**Fig. 3 Latent state bias and computational learning models. a** Example state time-courses, for each session, from two subjects. R and L denote the two possible states (decoder outputs). Each line represents trial-by-trial decoded outputs smoothed with a moving average filter (span = 5 trials). **b** Individual traces of the extent of latent RL state bias throughout each session. Latent state bias was defined as the unsigned difference between the number of occurrences of each state, normalized by the number of trials. Time-courses were computed with a moving window (span = 30 trials), and then smoothed with a moving average filter (span = 5 trials). The black dotted lines indicate the group mean, the shaded areas the 95% confidence intervals, coloured lines individual subjects. **c** Individual and group average of the degree of absolute latent state bias as outputted by the decoder, for each session. **d** Mean latent state bias plotted vs. mean $p$(opt action), averaged over sessions 1–2. Pearson correlation ($n = 18$), two-sided $p$ value. **e** Example time-courses of actions selected by two subjects (blue lines) vs. actions selected by the two RL algorithms that were fitted to the data. Top, black lines: state-free RL model. Bottom, grey lines: noisy state-dependent RL model. **f** Akaike Information Criteria (AIC)[31] was computed for each subject, session and model, respectively. Lower AIC indicates a better fit. Left: bars show the difference in AIC between the two models: $AIC_{sd} - AIC_{sf}$. $\Delta AIC < 0$ on all sessions, indicating lower AIC for the noisy state-dependent RL. Right: subject level and median $\Delta AIC$ for each session. Within-session statistical test against 0: full linear model, with the intercept as difference from 0 (two-sided $p$ values, FDR corrected). Between-session statistical test of difference (sessions 1 vs. 2): sign test (one-sided $p$ value, uncorrected). In **c**, **d**, **f**, coloured dots represent individual subjects; in **c**, **f**, thick horizontal lines represent the mean and the median, respectively, error bars the SEM. The experiment was conducted once ($n = 18$ biologically independent samples). **f** n.s.$P = 0.42$, **$P = 0.0085$, ***$P = 0.0014$.

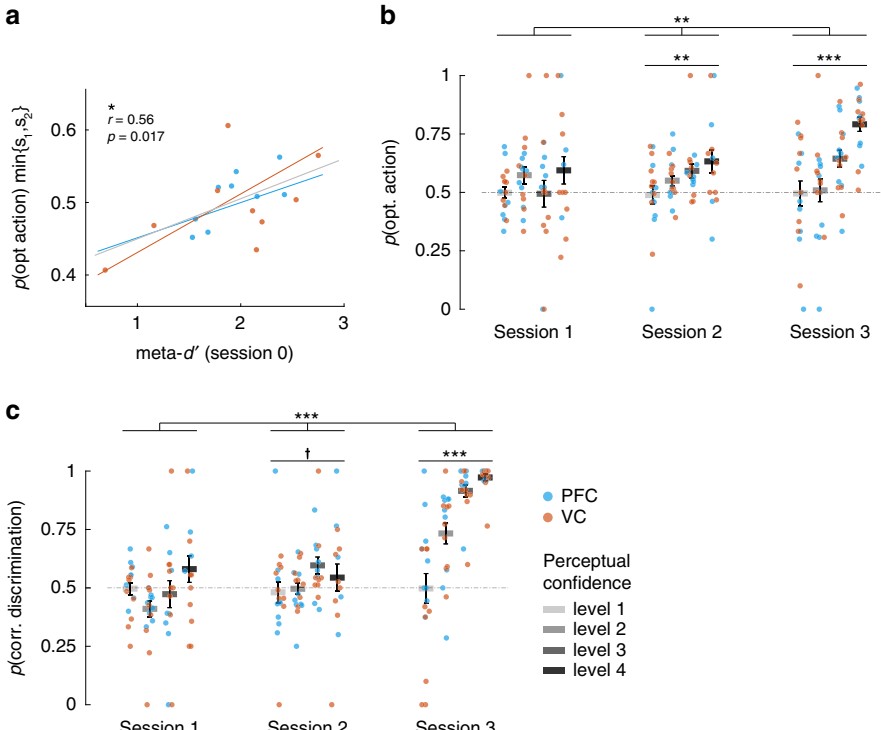

**Fig. 4 Metacognition correlates with learning to use latent brain activity. a** Across-subject correlation between the baseline (minimal) gambling performance attained in sessions 1 and 2, and individual metacognitive ability (how well one's confidence tracks discrimination accuracy). Metacognitive ability was computed with independent behavioural data from the decoder construction session (session 0, see 'Methods'). Pearson correlation ($n = 18$), two-sided $p$ value. **b** Proportion of optimal actions plotted by confidence level. The performance was measured as the proportion of trials in which the subject chose the action more likely to be rewarded, given the latent state. Significance was assessed with linear mixed effects models (two-sided $p$ values, uncorrected). **c** Discrimination accuracy, as leftward vs. rightward motion discrimination, plotted by confidence level. The correctness of the response was based on the output of the decoder. Significance was assessed with linear mixed effects models (two-sided $p$ values, uncorrected). For all plots, coloured dots represent individual subjects, grey bars the mean, error bars the SEM. The experiment was conducted once ($n = 18$ biologically independent samples). **b** **(interaction) $P = 0.0017$, **(session 2) $P = 0.004$, ***(session 3) $P = 4.68 \times 10^{-7}$; **c** ***(interaction) $P = 4.68 \times 10^{-6}$, †(session 2) $P = 0.078$, *** (session 3) $P = 7.03 \times 10^{-14}$.

discrimination is around chance level, decision confidence is hypothesized to correlate with learning reward associations. Although task accuracy and confidence judgements are usually highly correlated, it is possible, however, to uncover dissociations under several circumstances[16,32,33]. Importantly, previous work has shown that humans can track their task performance even when they claim to be unaware of the stimuli[28,29]. Besides, confidence has been associated with RL in the context of perceptual decisions, as a putative feedback channel[34,35]. We thus hypothesized a correlation between confidence and RL measures, reflecting the strength of learning, even while the relevant RL state information remained below consciousness.

We first quantified metacognitive ability, meta-$d'$[36], using independent data from the initial decoder construction stage (session 0, see 'Methods'). Roughly, meta-$d'$ estimates the trial-by-trial correspondence between confidence judgements and discrimination accuracy. In accordance with the hypothesis that metacognition could predict RL performance, we established that meta-$d'$ predicted the baseline $p$(opt action) attained within the first two unconscious sessions ($N = 18$, Pearson $r = 0.56$, $P = 0.017$; robust regression: $\beta = 0.057$, $t_{16} = 2.84$, $P = 0.012$, Fig. 4a). More metacognitive individuals had a higher starting baseline in the gambling task. Taking the two groups in isolation, this effect held for the PFC group ($N = 9$, Pearson $r = 0.72$, $P = 0.029$), but not VC group ($N = 9$, Pearson $r = 0.51$, $P = 0.16$), albeit the difference was not significant (one-sided $z$-test, $z = 0.60$, $P = 0.28$). Next, we found that the probabilities of optimal action-

selection increased with higher confidence from session 2 (Fig. 4b, LME model, data from all sessions, interaction between fixed effects 'session' and 'confidence' $\beta = 0.041$, $t_{194} = 3.18$, $P = 0.0017$; data restricted to session 1, $\beta = 0.018$, $t_{62} = 0.87$, $P = 0.39$; session 2: $\beta = 0.047$, $t_{62} = 2.98$, $P = 0.0041$; session 3: $\beta = 0.10$, $t_{68} = 5.57$, $P < 10^{-5}$, Supplementary Table 3). This result was further supported by confidence-related differences in discrimination rates (Fig. 4c, Supplementary Fig. 5a, b, Supplementary Table 4) and to the extent that subject level strength of confidence being predictive of optimal action rate correlated with the same effect in perceptual discrimination (Supplementary Fig. 5c).

One concern is that this pattern of findings may have arisen randomly or may have been triggered by an increase in confidence over time because of reward bias. However, in sessions 1 and 2 confidence was not different in trials that followed a reward compared with trials that followed the absence of reward (Supplementary Fig. 5d). A yoked control experiment in which new naive subjects received trial sequences from the main experiment did not reproduce these associations between confidence and action selection, nor a difference in confidence between sessions (Supplementary Fig. 6).

We next assessed the effect of confidence on RL$_{sd}$ with further computational analyses. Toward this end, we estimated the trial-by-trial magnitude of reward-prediction error (unsigned RPE, or | RPE|), which reflects the degree of uncertainty in learning[37]. Of note, the main assumption for this analysis is that the RL process (at least before session 3) happens below consciousness. But if the

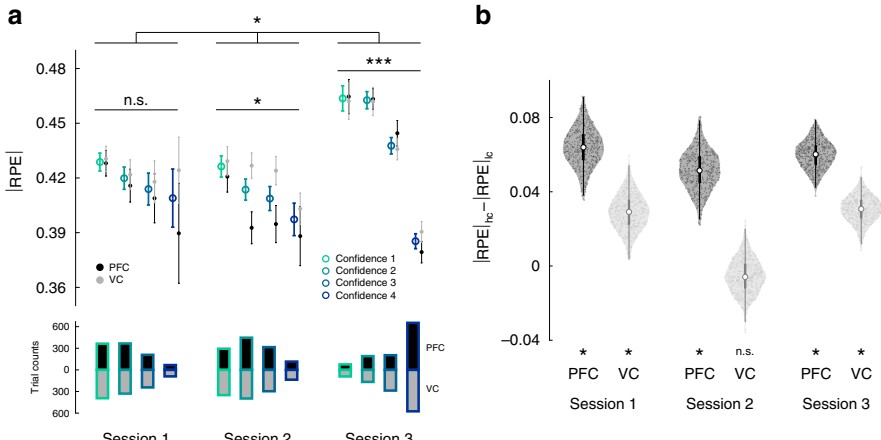

**Fig. 5 Computational modelling of behaviour: metacognition correlates with RL$_{sd}$.** Using the noiseless state-dependent RL algorithm we computed the magnitude of reward-prediction error |RPE|, which reflects the uncertainty in learning the gambling task. |RPE| and confidence ratings were taken from within the same trial. In temporal order this means that confidence was first, and |RPE| followed, as it was computed at outcome time (end of the trial). **a** |RPE| was significantly modulated by confidence from session 2: Higher perceptual confidence was associated with smaller |RPE|, meaning that a high confidence choice had lower probability to result in an unexpected reward. Coloured circles represent the median across all subjects pooled, light grey circles represent the median across subjects pooled from the VC group, and dark from the PFC group; error bars the SEM. The histograms at the base represent the trial counts for each confidence level, for both VC (lower) and PFC (upper) groups. Significance was assessed with linear mixed effects models (two-sided $p$ values, uncorrected). *(interaction) $P = 0.02$, n.s.(session 1) $P = 0.68$, *(session 2) $P = 0.012$, ***(session 3) $P = 1.56 \times 10^{-12}$. **b** Bootstrapped mean |RPE| difference between high and low confidence. Trials were first split according to the within-session median confidence: below the median as 'low confidence', equal or above the median as 'high confidence'. For each group, session, bootstrapping ($n = 500$ runs) was applied as follows: in each run, high and low confidence trials were sampled with replacement, and the difference of the means was thus computed. For each distribution, the absence of 0 in the 95% CI was taken as significance (*) at $P < 0.05$ (session 1: PFC = [0.045 0.083], VC = [0.010 0.046]; session 2: PFC = [0.032 0.073], VC = [−0.022 0.012]; session 3: PFC = [0.046 0.073], VC = [0.016 0.044]). The shaded areas in the violin plots represent the population spread and variance, the white dot the median, the thicker line the interquartile range, coloured dots individual bootstrap samples. The experiment was conducted once ($n = 18$ biologically independent samples); single-trial data were pooled across subjects.

brain has to learn some form of mapping between states (patterns of activity) and actions, then it should also store an approximation of the expected value of RL state-actions pairs (defined by the decoder output [state] and selected A or B [action]). Therefore, for this and all following analyses we utilized the RL$_{sd}$ model without noise to get an unbiased estimate of learning uncertainty. |RPE| traces were analysed with LME models (Supplementary Table 5). In order to provide a visual rendering, |RPE| was binned by confidence level (Fig. 5a): this revealed the existence of a coupling between |RPE| and confidence, with high confidence associated with low |RPE| and vice versa, low confidence with higher |RPE| (LME model, data from all sessions, significant fixed effect 'session': $\beta = 0.019$, $t_{6649} = 2.43$, $P = 0.015$; interaction between fixed effects 'group' and 'confidence': $\beta = -0.028$, $t_{6649} = 2.89$, $P = 0.004$; and interaction between fixed effects 'session', 'group' and 'confidence': $\beta = 0.0095$, $t_{6649} = 2.32$, $P = 0.02$; data restricted to session 1, fixed effect 'confidence': $\beta = 0.001$, $t_{2059} = 0.41$, $P = 0.68$; session 2: $\beta = -0.007$, $t_{2348} = -2.51$, $P = 0.012$; session 3: $\beta = -0.019$, $t_{2241} = -7.11$, $P < 10^{-3}$; full results in Supplementary Table 5). To further examine group differences, we split trials into low and high confidence bins (within-session median confidence split: trials with ratings below the median were labelled as 'low', those with ratings equal or above the median as 'high'). The coarser confidence partition further supported the larger effect size of confidence in the group with PFC decoder compared with the VC group (Fig. 5b). This finding raises intriguing questions on the function of metacognition and supports the view that its neural substrates are linked to prefrontal subregions[14,16,32].

**Neural substrates of learning and confidence–RL interaction.** In terms of RL, the most difficult element in this task is not for an

agent to predict the value for a given [state, action] pair per se (which would be trivial once the state is known), but rather to develop a closer estimate of the latent RL state itself (defined by a pattern of neural activity). At the onset of learning, several cortico-basal loops are predicted to be activated in a parallel search for the relevant (latent) states[24,38], alongside activity in the basal ganglia[39], with specialized encoding of multiple task and behavioural variables by dopamine neurons[40]. As RL progresses, the brain should use RPE to automatically select a few, relevant loops related to the latent RL states. Recent evidence indicates that RPE correlates change dynamically over time[38,41]. Here, using raw, signed RPE as a parametric regressor in a general linear model (GLM) analysis of fMRI signals, we found evidence that the brain initially undergoes a global search, spanning the anterior insula, anterior cingulate cortex, PFC subregions including DLPFC and ventromedial PFC, as well as the thalamus and basal ganglia (sessions 1–2, Supplementary Fig. 7). In session 3, RPE correlates were mainly restricted to basal ganglia, in line with classic theories of RL in neuroscience[39,42] (Supplementary Fig. 7). The differences in neural correlates between sessions 1–2 and 3 may be ascribed to differences in task properties. Alternatively, and more intriguingly, these could also reflect a convergence in the global search for task-states driven through RPEs. Correlates in the anterior cingulate cortex in sessions 1–2 can be linked to the intensive action-selection search, model updating and confidence evaluation that underpins learning under uncertainty[43–45]. The same analysis was repeated with $z$-scored RPEs (across subjects and sessions)[46], yielding comparatively similar results (Supplementary Fig. 8).

Resting-state functional connectivity is believed to be modified by recent co-activation of two brain areas and acquisition of knowledge or skills[47–49]. Given the nature of the task employed here and the important role of RPE in driving learning, perhaps

connections between specific brain regions and the RPE-encoding basal ganglia may be strengthened. Resting-state scans were collected prior to the learning task in each session (see 'Methods'); the seed region for the analysis was defined as the voxels in the basal ganglia found to significantly correlate with RPE in session 3 (data independent of all resting-state scans, right inset in Supplementary Fig. 9). We focused on changes related to session 2 (after–before), because this was the single time point where subjects showed strong evidence of learning, but where the RL states were still latent, unconscious. Strikingly, basal ganglia had increased connectivity with the right medial frontal gyrus (MFG, part of the DLPFC) and inferior parietal lobule (IPL) (Supplementary Fig. 9), both regions linked to confidence judgements and reliability of sensory evidence[16,32,50,51].

Behavioural and computational analyses have shown that metacognition correlates with RL along multiple axes. In light of increased resting-state connectivity between RPE-encoding basal ganglia and MFG/IPL as well as previous research[16,34,39], DLPFC and basal ganglia emerge as the logical neural substrate for this interaction. The metacognitive process could interact with |RPE| so as for the brain to evaluate how close an estimation is to the real RL state[24]. From this perspective, as learning progresses, we should see two effects: (1) confidence becoming predictive of the internal neural evidence for the latent RL state; (2) neural representations of confidence and |RPE| should perhaps become more synchronized, as they work together to facilitate learning. But an alternative possibility is that neural representations of confidence and |RPE| inform a putative downstream (region) state estimator to drive learning[52]. If that is the case, confidence could still correlate with the neural occurrence of the latent state, but the neural representations of confidence and |RPE| may not because their computations would unfold into independent processes.

We first found evidence for effect (1): confidence ratings correlated with the trial-by-trial fMRI multivoxel distance from the decoder classification boundary defining the task's latent RL states (Fig. 6a). That is, the greater the evidence in favour of one RL state, the higher the confidence. Importantly, this correlation measure increased during stimulus presentation, before perceptual decisions, confirming that confidence is retrieved explicitly only at report time, while it is likely computed earlier on. This suggests that perhaps metacognition could provide a means of accessing the artificial, low-dimensional manifold where classification boundaries are defined.

Second, we tested for effect (2) in the following manner. At the outset, we constructed a decoder for low vs. high confidence in the DLPFC, and a decoder for low vs. high |RPE| in the basal ganglia. By tabulating the outputs of the two decoders, $\chi^2$ statistics can be computed to quantify the degree of association (synchronization) between confidence and |RPE|. One thousand bootstrapped runs were calculated for each RL session: the distribution showed a marked shift towards higher $\chi^2$ values already from session 1 to session 2, then further increasing in session 3 (Fig. 6b). This implies that with learning, the independence of the two decoders' outputs decreased. That is to say, since these decoders based their predictions on patterns of voxels activity, that confidence and |RPE| representations became more coupled at the multivoxel level. The effect was specific for the pairs of interests (low confidence–high |RPE| and high confidence–low |RPE|, Fig. 6c). Consequently, the increase in resting-state functional connectivity between the DLPFC and the basal ganglia was coupled with increased synchronization of the information represented in the RL task, confidence and |RPE|. These results indicate that RL processes and cognitive modules actively interact during reward-based learning.

## Discussion

Two main questions were addressed in this study: Can human subjects learn to make use of latent, high-dimensional brain activity? What is the putative vehicle and neural substrate of this ability? The closed-loop design adopted here granted a unique opportunity to investigate the ability of the human brain to learn to use unconscious, high-dimensional internal representations. We show that hallmarks of learning emerge within a limited number of trials and errors, without explicit presentation of the relevant knowledge, and that initial metacognitive ability predicts subsequent task performance. We report here on a possible mechanism implemented by the brain. We speculate that meta-cognition could be useful to explore latent states and form low-dimensional representations, particularly so when necessary to drive efficient RL. The ability to learn hidden features in high-dimensional spaces is supported by an initially activated, distributed, and parallel neural circuitry that largely involves the basal ganglia and PFC. Such circuitry provides the neuroanatomical basis for the interaction between metacognitive and RL modules. Previous studies have highlighted the functional relevance of parallel cortico-basal loops in terms of RL and cognition[53,54], as well as the role played by metacognition in RL[34,55]. Our results further suggest that metacognition may go beyond an internal feedback mechanism to the basal ganglia[34], and help RL processes efficiently extract 'task state' information[56]. Work in rodents has shown that dopamine release in the basal ganglia and PFC has dissociable dynamics—a broadcast signal for learning and local control for motivation[57]. It would be interesting to answer how confidence (metacognition by extension) may influence this balancing act in order to promote faster learning or allow better control.

Is metacognition really relevant to reward learning? Since this study is limited because correlational in nature, a simpler and perhaps more parsimonious alternative model is that confidence is related to RL, but merely so because it reflects or reads out a successful latent state search. For example, we found that learning uncertainty (|RPE|) seemed to (mildly) influence future confidence ratings (i.e., next trial's judgements, Supplementary Fig. 10), although quite noisily. While this interpretation RL → metacognition cannot be entirely ruled out, our results strongly suggest that confidence could be instrumental for efficient RL (i.e., Figs. 4a–c, 5 and 6a). First, besides general correlations between metacognition and RL at several levels, metacognitive ability evaluated independently 1 week prior predicted later RL performance. Second, confidence during learning was unaffected by the outcome in the previous trial (Supplementary Fig. 5d). In the present task confidence judgments happened earlier in time than action selection, forcing its explicit computation early on—this could have then been used to inform RL state search. A compelling addition to this argument is that subjects whose decoder was based in PFC, a strong candidate as metacognitive substrate[16,32,58], also displayed larger effect sizes in confidence–RL correlation measures (Figs. 4a and 5a, b). These results cast doubt on the view that confidence is merely reflecting the previous trial's reward, thereby lacking any function. The picture is probably more nuanced, as decision confidence and learning uncertainty likely evolve in parallel but also with reciprocal modulations. As is the case with attention[59] and memory[60], confidence and RL processes probably interact repeatedly in time, with specific directionalities and constraints that depend on the time (before or after action/outcome)[56], the type of outcome (win or loss)[61], and whether the association is forming below or above consciousness. Future studies could further dissect these aspects of learning.

If confidence (and metacognition by extension) is involved in learning from rewards, what is the underlying computational

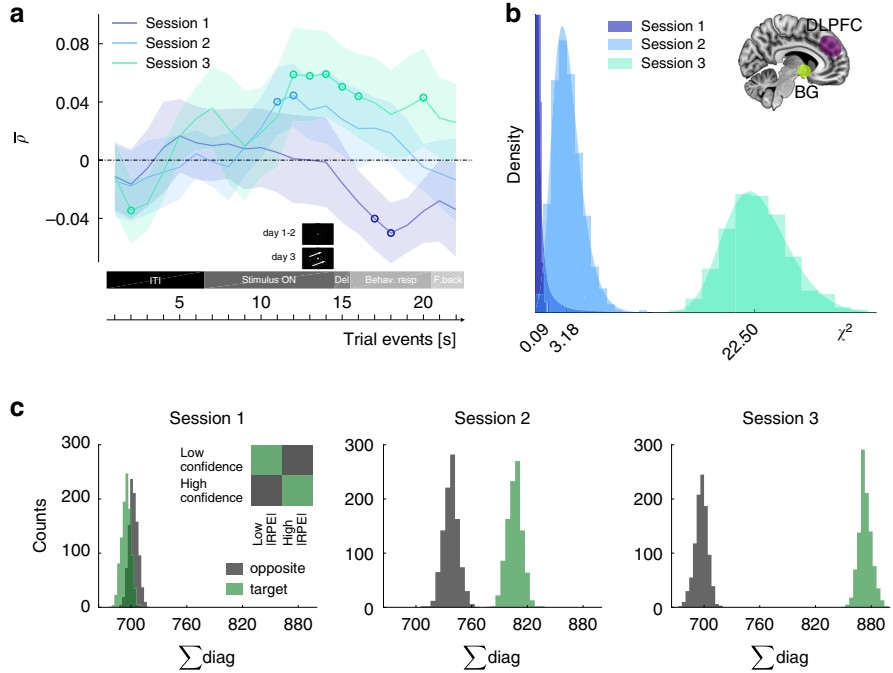

**Fig. 6 Correlations between confidence, latent state and reward-prediction error. a** Confidence judgements correlated with the amount of latent RL state evidence. Spearman rank correlation was computed for each subject between the trial-by-trial confidence ratings and the trial-by-trial dot product of decoder weights with voxels activities. Correlation coefficients were Fisher-transformed. Y-axis: $\rho$, coloured lines represent the group mean, shaded areas the SEM, circles the locations in time where the correlation was significantly different from zero (t-test against 0-mean, two-sided $p$ values, FDR corrected). **b** Multivoxel pattern association between basal ganglia and prefrontal cortex supporting confidence–|RPE| correlation. A decoder for confidence was built from multivoxel patterns in the DLPFC, while a decoder for |RPE| was built from multivoxel patterns in the basal ganglia. For each session, the original data were randomly resampled $N = 1000$ times at the subject level and then pooled across the population to create a $\chi^2$ distribution to indicate the degree of association between confidence and |RPE|. The distributions are plotted as histograms, overlaid with a shaded area generated from a standard generalized extreme value fit. **c** The histograms are based on the same distributions as in **b**, displaying here the sum of the occurrences of predicted confidence paired with predicted |RPE|. Target (green colour) is the sum of the occurrences where predicted high confidence paired with predicted low |RPE| (and vice versa). Opposite (dark grey colour) denotes the reverse pairing pattern (e.g., predicted high confidence−predicted high |RPE|). In **b**, **c**, $N = 1000$ resampling runs per subject, then pooled across the population. The experiment was conducted once ($n = 18$ biologically independent samples).

mechanism? Previous work in humans and rodents suggests that sensory confidence relates to uncertainty about the expected value of choices[51] and is combined with reward history[56]; this could in turn orchestrate a more fine-grained learning strategy and behavioural responses. An additional, thought-provoking possibility is that metacognition may also support efficient RL processes by enabling low-dimensional meta-representations in the PFC—similar to the 'chunking' phenomenon in working memory[62]. This way, RL processes could operate in a reduced state space, therefore weakening the major obstacle in terms of learning posed by the curse of dimensionality, the incommensurate computations needed for high-dimensional state spaces.

There are several limitations to the current study. First of all, the task does not have any experimental manipulation of the variables of interest (performance or confidence). Because of this, we have to rely on indirect evidence to rule on the directionality of the correlation between performance and confidence exposed here. Yet, by acknowledging this limitation, our design engenders one essential aspect: as experimenters we do not have to impose all conditions on the task, representations in the brain itself can be used to define the task spaces. As such, this design allows genuinely high-dimensional and unconscious information to be used in a specific manner, rather than by means of masked and/or very weak/noisy stimuli.

Subconsciousness in this study can be referred to the following three aspects, which are interrelated but not identical. (1) Unawareness of RL strategy, which was ascertained in post-experiment questionnaires, at least until session 2. (2) Unawareness about

activation patterns utilized by the decoder—subjects did not know about the closed-loop aspect of the task until post-experiment briefing at the end of session 3. Past experiments using a similar approach, where activity patterns are detected online with a decoder, found that in >97% of the cases subjects were unaware of the content and purpose of the manipulation[17]. (3) Chance-level discrimination accuracy about latent state (motion direction). To note, we found a trend that better discrimination accuracy was associated with better gambling performance, but this alone does not invalidate the claim that learning happened below consciousness, since a correct discrimination is important for the subsequent gambling action as both are based on the same latent state. That is to say, rewards in the gambling task could have evoked partial learning in the discrimination choices[7].

Although the task was based on stochastic representations captured by our decoders, one could always argue that, in principle, it was simple. We highlight here that, without knowing how the RL states were defined, this remained a complex, multidimensional problem for the brain—given the number of neurons (and voxels). Subjects did not know the location (PFC or VC) or sparsity of the voxels selected by the machine learning decoders, or the task time points used for real-time decoding. The imperfect classification accuracy (around 70%) also contributed to the inherent uncertainty in the brain's estimation of RL states (see Supplementary Note 2 for a more in-depth discussion on these points). Although visual direction information utilized here is simpler than cognitive/abstract thoughts, the problem in this task remains high dimensional. The information detected through

decoding in our task is probably closer to activity that arises during spontaneous thoughts/behaviours which shows richer activity patterns[63] (e.g., Supplementary Fig. 11). For the brain, solving this kind of problem is not trivial. It essentially has to pair implicit patterns of neural activity (which vary from trial-to-trial and are high-dimensional) to actions and rewards obtained after a delay. In order to learn quickly the brain has to operate at a more abstract level than sensory features; that is to say, reduce the dimensionality of the problem. We suggest metacognition is part of this mechanism. In fact, synchronization of neurons through electrical coupling or synchronization between brain areas via cognitive functions have been proposed as neural mechanisms controlling degrees-of-freedom in learning[24,64,65]. Metacognition and consciousness could thus have a clear computational role in adaptive behaviour and learning[25,66].

How do these findings integrate within the bigger picture of artificially intelligence (AI) and neuroscience? It is beyond the current scope to provide an explicit implementation of how metacognition and RL may interact at the neural level. Nevertheless, this is the first step in a direction we envision to be of some importance. In particular, work towards endowing artificial agents with self-monitoring capacities or the ability to operate at different representational levels (feature level, concept level, etc.) may bridge the gap between human and AI performances in real-world scenarios, beyond pattern-recognition problems[25]. Neuroscience-based principles such as the ones presented here can provide seeds to develop cognitively inspired AI algorithms[67] and is becoming a core aspect of work at the boundary between neuroscience and machine learning. Finally, the approach and the results discussed here may provide new ideas to investigate the functions of metacognition and the depth of unconscious learning in humans and animals.

## Methods

**Subjects**. Twenty-two subjects (mean 23.6 y.o., SD 4.0; 5 females) with normal or corrected-to-normal vision participated in stage 1 (motion decoder construction). One subject was removed because of corrupted data; one subject withdrew from the experiment after stage 1. We initially selected 20 subjects, of which one was removed after the first session of RL training due to a technical issue (scanner misalignment between stage 1 and new sessions), while a second subject was removed due to a bias issue with online decoding (all outputs were strictly of the same class). Thus, 18 subjects (mean 23.4 y.o., SD 3.3, 5 females) attended all RL training sessions. All results presented are from the 18 subjects who completed the whole experimental timeline, with a total of 72 scanning sessions.

All experiments and data analyses were conducted at the Advanced Telecommunications Research Institute International (ATR). The study was approved by the Institutional Review Board of ATR. All subjects gave written informed consent.

**Stage 1 (session 0) behavioural task**. The initial decoder construction took place within a single session. Subjects engaged in a simple perceptual decision making task[16]: upon presentation of an RDM stimulus they were asked to make a choice on the direction of motion and then rate their confidence about their decision (Supplementary Fig. 1). The choice could be either right or left, and confidence was rated on a 4-point scale (from 1 to 4), with 1 being the lowest level—pure guess, and 4 the highest level—full certainty.

The coherence level of the RDM stimuli was defined as the percentage of dots moving in a specified direction (left or right). Half of the trials had high motion coherence (coh = 50%). The latter half had threshold coherence (between 5 and 10%). On those threshold trials, coherence was individually adjusted at the end of a block if the task accuracy at perceptual threshold, ~75% correct, was not maintained.

The entire stage 1 session consisted of 10 blocks. A 1-min rest period was provided between each block upon the subject's request. Each block consisted of 20 task trials, with a 6 s fixation period before the first trial and a 6 s delay at the end of the block (1 run = 292 s). Throughout the task, subjects were asked to fixate on a white cross (size 0.5 deg) presented at the centre of the display. Each trial started with an RDM stimulus presented for 2 s, followed by a delay period of 4 s. Three seconds were then allotted for behavioural responses (direction discrimination 1.5 s, confidence rating 1.5 s). Lastly, a trial ended with an intertrial interval (ITI) of variable length (between 3 and 6 s).

Because subjects were in the MR scanner while performing the behavioural task, they were instructed to use their dominant hand to press buttons on a diamond-shaped response pad. Concordance between responses and buttons was indicated on

the display and, importantly, randomly changed across trials to avoid motor preparation confounds (i.e., associating a given response with a specific button press).

**fMRI scans acquisition and protocol**. The purpose of the fMRI scans in stage 1 was to obtain fMRI signals corresponding to viewed or perceived direction of motion (e.g., rightward and leftward motion) to compute the parameters for the decoders used in stage 2, the online RL training. All scanning sessions took place in a 3 T MRI scanner (Siemens, Prisma) with a 64-channel head coil in the ATR Brain Activation Imaging Centre. Gradient T2*-weighted EPI (echoplanar) functional images with blood-oxygen-level-dependent (BOLD)-sensitive contrast and multi-band acceleration factor 6 were acquired. Imaging parameters: 72 contiguous slices (TR = 1 s, TE = 30 ms, flip angle = 60 deg, voxel size = $2 \times 2 \times 2$ mm$^3$, 0 mm slice gap) oriented parallel to the AC–PC plane were acquired, covering the entire brain. T1-weighted images (MP-RAGE; 256 slices, TR = 2 s, TE = 26 ms, flip angle = 80 deg, voxel size = $1 \times 1 \times 1$ mm$^3$, 0 mm slice gap) were also acquired at the end of stage 1. The scanner was realigned to subjects' head orientations with the same parameters on all sessions.

**fMRI scans preprocessing for decoding**. The fMRI data for the initial 6 s of each run were discarded due to possible unsaturated T1 effects. The fMRI signals in native space were preprocessed in MATLAB Version 7.13 (R2011b) (MathWorks) with the mrVista software package for MATLAB [http://vistalab.stanford.edu/software/]. The mrVista package uses functions from the SPM suite [SPM12, http://www.fil.ion.ucl.ac.uk/spm/]. All functional images underwent three-dimensional (3D) motion correction. No spatial or temporal smoothing was applied. Rigid-body transformations were performed to align the functional images to the structural image for each subject. A grey-matter mask was used to extract fMRI data only from grey-matter voxels for further analyses. Regions of interest (ROIs) were anatomically defined through cortical reconstruction and volumetric segmentation using the Freesurfer software, which is documented and freely available for download online [http://surfer.nmr.mgh.harvard.edu/]. Furthermore, VC sub-regions V1, V2, and V3 were also automatically defined based on a probabilistic map atlas[68]. Once ROIs were individually identified, time-courses of BOLD signal intensities were extracted from each voxel in each ROI and shifted by 6 s to account for the haemodynamic delay using the MATLAB software. A linear trend was removed from the time-courses, and further z-score normalized for each voxel in each block to minimize baseline differences across blocks. The data samples for computing the motion (and confidence) decoders were created by averaging the BOLD signal intensities of each voxel for six volumes, corresponding to the 6 s from stimulus onset to response onset (Supplementary Fig. 1).

**Decoding multivoxel pattern analysis (MVPA)**. All MVP analyses followed the same procedure. We used sparse logistic regression (SLR)[69], which automatically selects the most relevant voxels for the classification problem, to construct binary decoders (motion: leftward vs. rightward motion; confidence: high vs. low; |RPE|: high vs. low).

K-fold cross-validation was used for each MVPA by repeatedly subdividing the dataset into a 'training set' and a 'test set' in order to evaluate the predictive power of the trained (fitted) model. The number of folds was automatically adjusted between $k = 9$ and $k = 11$ in order to be a (close) divisor of the number of samples in each dataset. Furthermore, SLR classification was optimized by using an iterative approach: in each fold of the cross-validation, the feature-selection process was repeated 10 times[70]. On each iteration, the selected features (voxels) were removed from the pattern vectors, and only features with unassigned weights were used for the next iteration. At the end of the k-fold cross-validation, the test accuracies were averaged for each iteration across folds, in order to evaluate the accuracy at each iteration. The number of iterations yielding the highest classification accuracy was used for the final computation, using the entire dataset to train the decoder that would be used in the closed-loop RL stage. Thus, each decoder resulted in a set of weights assigned to the selected voxels; these weights can be used to classify any new data sample.

Data from stage 1 (session 0) was used to train motion decoders. Pilot analyses indicated that the highest classification accuracies in PFC were attained by using high motion coherence trials alone (100 trials, 50 samples per class). Motion decoders were constructed with fMRI data from two brain regions: PFC and VC. These data were time-course extracted from the 6 s from stimulus onset to response onset. Because decoding motion direction always works better in VC—subjects were assigned to one group or the other (PFC or VC) so as to minimize the difference in overall classification accuracy between the groups to avoid further confounds arising simply because of different decodability. Overall, this meant that subjects with high classification accuracy in PFC were assigned to the PFC group, while those with low accuracy in the PFC were assigned to the VC group. See Supplementary Table 8 for subject-specific subregions. The mean (±SEM) number of voxels available for decoding was $3222 \pm 309$ for VC and $4443 \pm 782$ for PFC. The decoders selected on average $80 \pm 15$ voxels in VC and $63 \pm 18$ in PFC. The cross-validated test decoding accuracy (mean ± SEM) for classifying leftward vs. rightward motion was $70.44 \pm 2.63\%$ for VC and $65.51 \pm 1.35\%$ for PFC (two-sample t-test, $t_{16} = 1.67$, $P = 0.11$).

For confidence decoders, trials from stage 1 (session 0) with threshold coherence were used (100 trials) in order to avoid potential confounds due to large

differences in stimulus intensity. Because confidence judgments were given on a scale from 1 to 4, trials were first binarized into high and low confidence ratings, as described previously[16]. Confidence decoders were constructed with fMRI data from dorsolateral PFC (DLPFC, which included the inferior frontal sulcus, middle frontal gyrus, and middle frontal sulcus), and time-course extracted from the 6 s from stimulus onset to response onset. The mean (±SEM) number of voxels available for decoding was 6641 ± 183, and the decoders selected on average 40 ± 8 voxels. The cross-validated test decoding accuracy (mean ± SEM) for classifying high vs. low confidence was 68.77 ± 1.53%.

For RPE magnitude (unsigned RPE) decoders, fMRI data from stage 2 was used (see sections 'Stage 2 (session 1, 2, 3) online RL training' and 'RL modelling' for a description on the task, timing and computation of trial-by-trial RPE). All trials from session 3 were used and, similar to confidence decoders, trials were labelled according to a median split of the |RPE|. For example, if |RPE| was larger than the median, the associated trial was labelled as high |RPE|. |RPE| decoders were constructed with fMRI data from basal ganglia (which included bilateral caudate, putamen and pallidum), and time-course extracted from the 2 s from monetary outcome presentation. The mean (±SEM) number of voxels available for decoding was 3583 ± 81, and the decoders selected on average 69 ± 14 voxels. The cross-validated test decoding accuracy (mean ± SEM) for classifying high vs. low |RPE| was 57.34 ± 0.64 %.

**Stage 2 (session 1, 2, 3) online RL training**. Once a targeted motion decoder was constructed, subjects participated in three consecutive sessions of RL online training (Fig. 1). In the RL task, state information was directly computed from fMRI voxel activity patterns in real time. The setup allowed us to create a closed loop between (spontaneous) brain activity in specific areas and task conditions (behaviour). The loop was unknown to subjects; the only instruction they received was that they should learn to select one action among two options, in order to maximize their future reward.

On each session, subjects completed up to 12 fMRI blocks; on average (mean ± SEM) 9.9 ± 0.4, 11.2 ± 0.2 and 10.5 ± 0.2 blocks in session 1, 2 and 3, respectively. For some subjects ($n = 6$) one or more blocks (max 3 out of 12, in one case) had to be removed from subsequent fMRI analyses due to issues during real-time scanning. Nevertheless, whenever possible, these data points were used for behavioural analyses. Each fMRI block consisted of 12 trials (1 trial = 22 s) preceded by a 30-s fixation period and ending with an additional blank 6 s (1 block = 300 s). Furthermore, on each session, before the reinforcement task, subjects underwent an additional resting-state scan of the same duration (300 s).

The construction of an online trial observed the following rule. After a 6 s blank ITI (black screen), the RDM was presented for a total of 8 s. The first 6 s were always random (0% coherence), while in session 3 the last 2 s of RDM had coherent (coh) dot motion, computed as

$$coh = c \cdot \arctan(L - 0.5), \tag{1}$$

where $L$ is the likelihood, the output of the motion decoder; $c$ a constant, which increased over the first half of the experimental session following a sigmoid function over the interval (0 1). Negative values indicated leftward motion, while positive values rightward motion. This allowed us to have high coherence in the latter half of session 3. Additionally, the strength of the RDM stimulus was modulated by the contrast of the dots on a black background. Contrast was set at a fixed value of 20% in session 1 and session 2 while in session 3 it sigmoidally increased up to 100% over the first half of the experimental session, staying fixed thereafter. Importantly, because the operation of stimulus presentation and online decoding were performed by two parallel scripts on the same machine, the stimulus was presented in brief intervals of dot motion lasting 850 ms, followed by a short blank period of 150 ms. The presence of the blank period allowed the two processes to communicate in order to compute the new coherence level from the decoder output likelihood. Although this was effectively carried out only in session 3, the same design was used on each session for consistency between sessions. Following RDM presentation and a 1 s blank ITI, subjects had 1.5 s to make a discrimination choice (choose leftward or rightward motion), and 1.5 s to give a confidence judgement on their decision (on a scale from 1 to 4). Lastly, subjects had to select one of two actions, A or B, in order to maximize their future reward. The reward rule for options A and B was probabilistic and determined by the decoded brain activity. Each option was thus optimal only in one state (e.g., A when left motion was decoded from multivoxel patterns, B with right motion). The probability of receiving a reward was ~80% if the choice was congruent with the rule, ~20% otherwise. A rewarded trial corresponded to a single bonus of 30 JPY. On each session, up to 3000 JPY could be paid in bonus to a subject. Crucially, the reward association rule and the presence of online decoding were withheld from subjects: they were simply instructed to explore and try to learn the rule that would maximize their reward.

Because brain activity patterns alone were defining whether a trial was to be labelled as rightward or leftward—the experimenter had no control over the occurrence of either state (leftward or rightward motion representation). Behavioural responses could not be associated with a specific button press: pairings between buttons and responses were randomly determined on each trial and cued on the screen during response times.

**Real-time fMRI preprocessing**. In each block, the initial 10 s of fMRI data were discarded to avoid unsaturated T1 effects. First, measured whole-brain functional images underwent 3D motion correction using Turbo BrainVoyager (Brain Innovation). Second, time-courses of BOLD signal intensities were extracted from each of the voxels identified in the decoder analysis for the target ROI (either VC or PFC). Third, the time-course was detrended (removal of linear trend), and z-score normalized for each voxel using BOLD signal intensities measured up to the last point. Fourth, the data sample to calculate the RL state and its likelihood was created by taking the BOLD signal intensities of each voxel over 3 s (3TRs) from RDM onset. Finally, the likelihood of each motion direction being represented in the multivoxel activity pattern was calculated from the data sample using the weights of the previously constructed motion decoder. The final prediction was given by the average of the three likelihoods computed from the three data points.

**RL modelling**. We used a standard RL model[4,71] to derive individual estimates of how subjects' action selection was dependent on past reward history tied to actions and states (state-dependent RL: RL$_{sd}$) or actions alone (state-free RL: RL$_{sf}$). RL$_{sd}$ and RL$_{sf}$ are formally described as:

$$Q(s,a) \leftarrow Q(s,a) + \alpha \cdot (r - Q(s,a)), \tag{2}$$

$$Q(a) \leftarrow Q(a) + \alpha \cdot (r - Q(a)), \tag{3}$$

where $Q(s,a)$ in (2), $Q(a)$ in (3), is the value of selecting A or B. The value of the action selected on the current trial is updated based on the difference between the expected value and the actual outcome (reward or no reward). This difference is called the RPE. The degree to which this update affects the expected value depends on the learning parameter $\alpha$. The larger $\alpha$, the more recent outcomes will have a strong impact. On the contrary, a small $\alpha$ means recent outcomes will have little effect. Only the value of the selected action (which is state-contingent in (2)) is updated. The values of the two actions are combined to compute the probability $P$ of predicting each outcome using a softmax (logistic) choice rule:

$$P_{s_i,A} = \frac{1}{1 + e^{-\beta(Q(s_i,A) - Q(s_i,B))}}, \tag{4}$$

$$P_A = \frac{1}{1 + e^{-\beta(Q(A) - Q(B))}}, \tag{5}$$

The inverse temperature $\beta$ controls how much the difference between the two predictions values for A and B influences choices.

We used a noisy version of the RL$_{sd}$ (2) because this is a much more plausible scenario: this model assumes that access to the state information is partial, and stochastic. Noise was implemented by allowing the $Q$-value to be updated on the alternative state rather than the real state (as defined by the decoder output) on a subset of trials. Because of the stochastic nature of the process, we evaluated the model over 100 resampling runs, each with 100 noise levels—from 0 to 50%. The optimal level of noise—that is, leading to the highest log-likelihood—was determined by averaging the log-likelihood for each noise level over all resampling runs and then taking the maximum.

Furthermore, the two hyperparameters $\alpha$ and $\beta$ were estimated by minimizing the negative log-likelihood of choices given the estimated probability $P$ of each choice. We conducted a grid search over the parameter spaces $\alpha \in (0, 1)$ and $\beta \in (0, 20)$ with 50 steps each. The fitting procedure was repeated for each subject and each session (see Supplementary Table 9, group mean ± SE). For model comparison, RL$_{sd}$ had $k = 3$ parameters, while RL$_{sf}$ had $k = 2$. Trial-by-trial RPE measures were computed for each RL model, subject and session by fitting the data with the estimated parameters. RPEs were then used as inputs for offline analyses as described below.

**RPE-based analyses parametric GLM**. Image analysis was performed with SPM12 [http://www.fil.ion.ucl.ac.uk/spm/]. Raw functional images underwent realignment to the first image of each session. Structural images were re-registered to mean EPI images and segmented into grey and white matter. The segmentation parameters were then used to normalize and bias-correct the functional images. Normalized images were smoothed using a Gaussian kernel of 7 mm full-width at half-maximum.

Onset regressors at the beginning of outcome presentation were modulated by a parametric regressor, trial-by-trial RPE from RL$_{sd}$. Other regressors of no interest included regressors for each trial event (RDM, choice, confidence, action selection), motion regressors (6) and block regressors. The GLM analysis was repeated twice, once with raw RPE and once with z-scored RPE (across sessions and subjects)[46].

Second-level group contrasts from the GLM were calculated as one-sample $t$-tests against zero for each first-level linear contrast. Activities were reported at a threshold level of $P(\text{FPR}) < 0.001$ ($z > 3.1$, false-positive control meaning of cluster forming threshold). Statistical maps were projected onto a canonical MNI template with MRIcroGL [www.nitrc.org/projects/mricrogl].

**Connectivity analyses**. At the beginning of each session resting-state data were acquired during a window of 6 min. For connectivity analyses of resting-state data, we used the CONN toolbox v.17 [www.nitrc.org/projects/conn, RRID: SCR_009550]. Briefly, resting-state data underwent realignment and unwarping,

centred to (0,0,0) coordinates, slice-timing correction, outlier detection, smoothing and finally denoising. At the first level, we performed a seed-based correlation analysis, testing for significant correlations between voxels in a seed region and the rest of the brain. The seed was defined as the cluster of voxels within the basal ganglia that best tracked the RPE fluctuations on the last session of the RL task (session 3, independent data). The analysis was repeated for each session of resting-state scanning (session 1, 2, 3). Second-level results were calculated as one-sample $t$-tests against zero for each first-level contrast. We focused the second-level analysis on the two resting-state scans before and after RL session 2. We tested for the presence of the main effect, including all subjects at once, reporting between sessions contrasts (after > before) at a height threshold of $p < 0.001$ ($t > 3.65$, uncorrected), and $P(FDR) < 0.05$ for cluster size. Statistical maps were projected onto a canonical MNI template with MRIcroGL.

**Statistical analyses with LME models**. All statistical analyses were performed with MATLAB Version 9.1 (R2018b) (MathWorks), both with built-in functions as well as with functions commonly available on the MathWorks online repository or custom-written code. Effects of learning on behavioural data over several sessions and additional effects were statistically assessed using LME models with the MATLAB function 'fitglme'. Post hoc tests included LME over single sessions, restricted to certain variables as well as $t$-tests.

To evaluate the effect of confidence (levels from 1 to 4), session (1–3) and group (PFC, VC) on the dependent variable $y$ (I: probability of selecting optimal action, II: perceptual discrimination, III: |RPE| from RL$_{sd}$), we used the general model (in Wilkinson notation): $y \sim 1 + group \times session \times confidence + (1|subjects)$, which included random effects (intercept) for each subject, and 8 fixed effects (intercept, group, session, confidence, group: session, group: confidence, session: confidence, group: session:confidence). Whereby a simpler model (i.e., without three-way interaction), $y \sim group \times session + group \times confidence + session \times confidence + (1|subjects)$ fit the data equally well (likelihood ratio [LR] test indicating no difference, at $P > 0.05$), results from the simpler model are reported (alongside with LR statistics). Where a significant effect of 'session' or interaction between fixed effects 'session' and 'confidence' and/or 'group' was found, post hoc tests were carried out on data restricted to single sessions. For single-session data the general model $y \sim group \times confidence + (1|subjects)$ was used; whereby a simpler model (i.e., without interaction) fit the data equally well, results from the simpler model are reported.

The same approach was used to evaluate the effect of |RPE| on confidence (|RPE| from trial-1): the same equations and procedure, defining $y$ as confidence, while |RPE| was treated as a fixed effect.

**Offline multivoxel pattern analyses (Fig. 6b, c)**. For each session of the RL task, we used the set of voxels selected by confidence (DLPFC) and |RPE| (basal ganglia) decoders (described in the 'Decoding multivoxel pattern analysis' section) to compute the degree of association between confidence and |RPE| at the multivoxel pattern level. For |RPE|, the dataset was composed of the predicted labels (high, low |RPE|) of all trials within a session. To issue these predicted labels, we inputted the preprocessed voxel activities during the 2 TRs corresponding to action-selection outcome to the |RPE| decoder. For confidence, the prediction was extended to several time points. Specifically, the search was extended to TRs 7–16 (TRs corresponding to stimulus presentation, as well as those showing high correlation between confidence and RL state in sessions 2 and 3). Within the range 7–16 TRs we took the averaged raw voxel activities over 3TRs for a better S/N ratio before inputting data to the confidence decoders. As such, we obtained nine predictions for each trial, and selected the single one leading to the highest association strength between confidence and |RPE| predictions over all trials, at the subject level. Finally, we obtained two vectors of the same length (number of trials within a session) of predicted |RPE| (high, low) and confidence (high, low). These vectors from each subject were concatenated and the final degree of association was thus computed through $\chi^2$ statistics. The process was repeated over 1000 resampling runs by changing the subset of trials used to compute the confidence predictions at the subject level. This allowed us to create a distribution of 1000 $\chi^2$ values reflecting the overall degree of association between multivoxel patterns predicting confidence in the DLPFC and |RPE| in the basal ganglia.

At the single-trial level, predicted data points were categorized according to the following labels: *target* if the prediction were high confidence–low |RPE| or low confidence–high |RPE|, and *opposite* if the predictions were high confidence–high |RPE| or low confidence–low |RPE|. For each resampling run we summed all occurrences of target and opposite, creating a distribution of 1000 values. Overlapping distributions means that there is no association.

**Reporting summary**. Further information on research design is available in the Nature Research Reporting Summary linked to this article.

## Data availability

All data used to generate the figures and results of this paper are freely available within a stand-alone computing capsule at Code Ocean [https://codeocean.com], with https://doi.org/10.24433/CO.8602350.v2. Additionally, the Source Data underlying Figs. 2-6 can be found within the capsule in the /data/ panel, under the following file names: summarydata.mat, behavdata_preproc.mat, metacog-ability.mat, Qlearn_models.mat, FIG_6A_data.mat, CHI_pconf_prpe.mat. A reporting summary for this Article is available as a Supplementary Information file.

## Code availability

Custom code used to generate the figures and results of this paper is freely available within a stand-alone computing capsule at Code Ocean [https://codeocean.com/], with https://doi.org/10.24433/CO.8602350.v2.

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

## Acknowledgements
We thank Kaori Nakamura and Yasuo Shimada for experimental assistance and Drs. Ben Seymour, Kenji Doya, Brian Odegaard, Brian Maniscalco, Vincent Taschereau-Dumouchel, Matthias Michel and Giuseppe Lisi for helpful comments on earlier versions of the manuscript. A.C and M.K. were supported by JST ERATO (Japan, grant number JPMJER1801) and by AMED (Japan, grant number JP18dm0307008). H.L. was supported by the National Institutes of Health (US, grant number R01NS088628).

## Author contributions
A.C., H.L. and M.K. contributed to the conceptual development of the work and designed the experiment, and wrote and revised the manuscript. A.C. collected and analysed the data, and drafted the manuscript.

## Competing interests
The authors declare no competing interests.
