## [Peer Review File · Nature Communications]

Reviewers' comments:

Reviewer #1 (Remarks to the Author):

This manuscript reports the results of an fMRI study in which a real-time neurofeedback loop is used to define states that are subsequently used to guide the reinforcement of actions. Essentially the state that a decoder retrieves from either VC or PFC about the direction of a random dot stimulus defines which action leads to reward with high probability (and in session 1 and 2 the decoded state is not explicitly made known to the participant). Participants learn to perform the task across two sessions, with performance better than chance by session 2, albeit not significantly different from session 1. The authors take this data and then use it to test a hypothesis about the role of metacognition in facilitating learning about the use of "latent and "high dimensional brain states". The authors recorded confidence ratings – and found that confidence ratings increased over the course of the experiment, which they also found was correlated with performance, while correct discrimination accuracy per se was correlated more weakly with performance. Leveraging neuroimaging correlations with RPE across the three sessions, the authors conclude that metacognition boosts RL performance via a frontostriatal mechanism by which the brain untangles the course of dimensionality.

While the finding that people are able to learn to perform a task such as this is an interesting result and a solid technical achievement, the main claims of the paper about the boosting relationship of meta-cognition on learning of RL task representations, is not well supported by the experimental design, or the findings. Essentially what the paper does report is a correlation between metacognitive judgement and learning capacity both across subjects, and within subjects across sessions, but it is unclear in what direction does that correlation go.

1/ The basic issue with this whole manuscript is that to address a question about the relationship between meta-cognitive judgments and reinforcement learning over latent states, there are many experimental designs that could be selected for doing so, but using a neural decoding approach is not one of them. A much simpler and more powerful design would involve giving people tasks with hidden latent states without any neural decoding loop, and then tracking the degree to which they are able to solve the task successfully by making inferences over the hidden states. Experimental manipulations could be used to increase or decrease confidence and increase or decrease discrimination accuracy – indeed with a clever design they could be independently manipulated. The relationship between those inferences and confidence judgements could be tracked and analyzed. There is absolutely no need to have the neurofeedback component, and the design is problematical because there is no explicit experimental manipulation of any of the relevant decision variables. As a result, we can't conclude whether confidence has any causal role in driving performance of the latent state inference, or not. Though these two variables are correlated across the experiment we don't know what direction is the causality – only that the two are related (performance on the task and confidence in that performance).

2/ The argument that participants are performing a search through a massively high dimensional space to solve the task does not seem credible. First of all, the same subjects completed a motion dot discrimination task a week before that was used to train the classifier – in which the discrimination was left vs right movement. In the task itself the participants are asked to indicate whether movement was left or right. So, the participants are clearly clued into the key states of the task that are likely to be relevant for learning: left vs right moving dots. They are not faced with an unconstrained multidimensional problem. Instead, it should be obvious (whether consciously or not) given the way the task is structured that they need to focus on movement of the dots being either leftward or rightward.

3/ None of the computational modeling results help to make the case stronger about a role for metacognition in boosting learning about latent state spaces. The computational model analyses do help to validate the RL model by showing that a model which can utilize the relevant latent spaces will account better for participants behavior than one which does not, as participants do learn the task by session 2. The modeling also provides further support of a relationship between confidence judgements and learning accuracy, as indexed by a model-derived variable of unsigned prediction errors. So overall, the modeling is a nice validation of the role of RL in describing performance on this task, and on the relationship between conscious ratings of confidence and performance. However, these modeling results don't make the case any stronger about whether or how metacognition is causally relevant for learning.

4/ The neuroimaging results unfortunately are not informative and don't provide specific insight into the claims made by the authors. The authors descriptively report differences between session 1 and 2 and session 3 in correlations with RPE signals, concluding that there is initially a "global search" involving multiple brain regions that converges to "reward" and "belief" processing hubs in PFC and basal ganglia by session 3. But session 3 is a different task to session 1 and 2 (in session 3 the latent states are revealed to the participant) so it cannot be compared in kind to task 1 and 2, and in any event there is no control for changes in brain activity that would simply happen as a function of time on task / habituation etc. Furthermore, we have no idea whether changes in resting state reported before vs after the task are anything to do with task performance, as opposed to reflecting changes that just happen after being in the scanner for a certain period of time.

5/ The sample size is very low (n=18) – this is especially the case because the decoder was trained on two different brain areas VC vs PFC in half of the subjects. The authors report no difference between the subjects in terms of where decoding was performed but with such a small sample size, this is not an interpretable result. The sample size does affect the strength of some of the claims – for instance the lack of a significant correlation between task performance and discrimination accuracy could be entirely due to the small sample size.

Reviewer #2 (Remarks to the Author):

I have reviewed this manuscript previously. I thought that it was novel and interesting then. It has now been further revised and improved. I have no further comments. I have included my previous comments below.

This is a rather beautiful study that deserves to be seen by a wide audience. The findings will be of broad interest to visual and cognitive neuroscientists and to researchers building neuroimaging analysis tools and brain-inspired neural networks. The manuscript presents evidence for a very interesting conceptual advance that although intuitive is novel – that metacognition can be employed to distinguish one's own brain states. The argument is based on both careful behavioural analysis and careful interrogation of neural activity measurements. The methods and analyses employed are advanced and compelling. I have noted a number of points below but they are all minor in nature.

1 P6 The difference between the state free and the state dependent learning models initially seemed to me to have been explained clearly in lines 147-153 and in the legend for figure 3e on lines 187-195. However, it did not understand why the state-free model predicts learning that "would just follow

any average bias of the multivoxel patterns to one state - that is, the prevalence of one latent state over the other, if any (figure 3A-B)? Nevertheless, this seems to be an important idea that underlies several of the analyses illustrated in figure 3. Is it because if learning is dependent on the action the subject makes given the brain state, $Q(s,a)$, rather than dependent on just the state, $Q(s)$, or just the action, $Q(a)$, then neither one state or the other is likely to predominate in the brain because it is not directly associated with reward? Instead what is associated with reward is the action given the brain state. Why, however, should there be any drift in the underlying brain state towards one value in the state free $Q(a)$ model?

2 Figure 3. What do the orange and blue dot colours indicate? At first I wondered if these indicate participants from the PFC or the V1 recording experiments. The legend does not seem to suggest this. It does mention "Coloured dots (light/dark green) represent single subjects" but I could not easily see any green dots.

3 Line 298 "Resting state functional connectivity is believed to be modified by recent co-activation of two brain areas". Direct evidence that this is the case is provided by Johnen et al., (eLife, 2015) who directly activated pairs of human brain areas and then recorded connectivity with fMRI.

4 Lines 308-312 These lines refer to the changes in connectivity with voxels in the basal ganglia that are illustrated in figure 6b. They emphasize connectivity with "putamen, thalamus and cerebellum subregions" but these changes are hard to discern on panel 6b. Maybe they could be labelled? Instead what is more striking is the increased connectivity with parts of prefrontal cortex and anterior temporal lobe and the decreases in connectivity with more posterior cortical regions (bottom right). However, this seems unmentioned in the text.

5 Line 312 group-specific changes in functional connectivity are described as being shown in figure S6 but I think that the relevant figure is S5. I think that the key analyses already performed are interesting and I understand why they have been seeded in the manner they have. Nevertheless, it might be possible to obtain stronger striatal connectivity group by seeding from a posterior striatal location in the posterior striatum which is more closely linked to representations of rewarding visual stimuli (eg work from Hikosaka's laboratory such as Kunimatsu et al., J. Neuroscience, 2019; Amita et al., EPN, 2019). I do not, however, think that this is a particularly critical analysis.

6 Figure 7c The target distributions are described as gold coloured in the legend but they do not look gold coloured on my pdf (more grey coloured).

Reviewer #3 (Remarks to the Author):

Cortese and colleagues investigated whether humans can be trained to make strategic use of latent representations in their own brains. To this aim, they scanned subjects performing a RL task, where the mapping between action and reward was determined by "decoded mental states" about random dot motion (the classifier being based on a previous scanning session). They further propose that stated confidence mediates dimensionality reduction across the otherwise vast search space of latent brain states. The topic is interesting, and the authors should be praised for developing and combining a lot of sophisticated analyses. However, I think the paper presents several major issues, that I am not sure can be addressed in a revision without a replication experiment.

1. First, I really can't see how the results and the analysis support the claim that "meta-cognition

facilitates the search in latent task representations". The authors' interpretation strongly relies on a causal link that goes from confidence reporting to latent space search, but as all their evidence is correlational, it is, at least, equally likely to suppose a parsimonious alternative model where stated confidence merely reflects (or reads out) a successful search. All claims about confidence "helping" searching in the latent space should be qualified as speculative, as it may be that successful search is actually "helping" confidence expression.

2. Second, and more worrying, virtually most of the main results of the paper are significant at a very permissive threshold (uncorrected $p \sim 0.05$ or $p \sim 0.01$), which poses a serious multiple comparison issue, considering the considerable number of statistical tests performed. The same statistical concerns apply to imaging results which are reported at $p < 0.005$ uncorrected. This is even more problematic, as the study is clearly exploratory and relies on non-standard analytical procedures. Just to be clear, I am not suggesting that exploratory and innovative research should not be performed (or highlighted by important journals), but I feel that the current amount statistical evidence is not convincing enough. Fortunately, there is a straightforward solution for this: run the same experiment in a replication sample and see whether the results hold.

3. Third, also concerning the statistical analyses, in several instances I noted a "difference in significance does not imply significant difference" issue (for example $p(\text{opt action})$ significant in session 2, but not in session 1 as a proof of learning). Please, test directly the difference and refrain from interpreting difference in significance. In addition, as several crucial measures require test against "chance level", full linear models, where the intercept can be taken as difference from chance, would be better.

4. Please make clearer what "perceptual discrimination" in session 1 and 2 means. Since random dot motion was random and the 'state' was determined the decoded from neural activity, it is not immediately clear to see what is an accurate response here. In general the paper could benefit from clearer definitions of the dependent variables.

5. The fact that there was a significant trend for a correlation between accuracy in the two tasks, is also problematic (especially because the difference between the two tasks is only significant at $p < 0.028\dots$). Discrimination accuracy should be entered explicitly as a control for $p(\text{opt action})$. I fear that if the authors regressed out the shared variance, the residual will show no sign of significant conditioning. This is also true for the analysis of accuracy as a function of the level of perceptual confidence, where the discrimination accuracy follows a very similar pattern.

6. To link the learning effect to meta-cognition, the authors start by showing a correlation between meta- d' and the sum of the rewards (to note, also marginal $p < 0.041\dots$). Furthermore, it is very unclear why here the metric "sum of the rewards" has been chosen as a proxy for performance in the RL task. The analysis should be replaced with $p(\text{opt action})$, which has been the main dependent variable all along. Also it is unclear whether this correlation holds for both the PFC and the VC. In any case counting a subject twice does not make any sense.

7. The authors regress RPE against brain activity (model-based fMRI), but see Lebreton et al. (Nature Human Behavior; 2019) for issues related to this technique. In particular the authors should specify whether the RPE regressors have been z-scored, across subjects and/or session, justify their approach and specify whether the results are affected by their analytical choices.

Minors

The authors equate reward contingencies to RL states. I see why this is the case in their design, but

different states may have the same reward contingencies and vice versa. Please consider avoid this terminology confusion.

Oddly, the result session starts with a statement about the interpretation of the results, before providing any relevant quantitative and statistical evidence. Please consider replacing this sentence with more factual information.

"implicit metacognition" is mentioned in the abstract, but not well defined. The definition given in the abstract is not clear for a scientist not expert in meta-cognition.

In the abstract there is mention of possible applications of these findings in artificial intelligence. This is a pretty wild speculation (the models proposed are very simple, computers do not have brain and the role of meta-cognition in AI is very much debated), so please remove it from the abstract.

The term "brain states" should be clarified at its first occurrence (in the instruction). Different neuroscientists will have very different interpretations of what a brain state is.

The term "discrimination confidence" should be clarified at its first occurrence (in the instruction). It is unclear for a scientist outside meta-cognition field.

POINT-BY-POINT RESPONSE TO THE REVIEWERS:

We thank the anonymous Reviewers for their thoughtful reviews and constructive comments. We find their reviews to be of great importance to ameliorate the quality of the manuscript. Here, we answer to the points that were raised.

Reviewer #1 (Remarks to the Author):

1/ The Reviewer points out that one cannot make causal claims unequivocally with observational data alone, as in the current task there is no explicit manipulation of the relevant variables. A different task design would have been more appropriate to address a question about the relationship between metacognition and reinforcement learning.

We are sorry for the confusion, but the main objective of the present study was actually to test if humans can learn a task in which the information that determines the RL states is (a) high dimensional and (b) unconscious. The goal was to assess whether such learning is at all possible under such extreme circumstances, but not the general relationship between metacognition and RL per se. The possibility of metacognition being causally involved was only brought up because the primary objective of the study was met - surprisingly we observed such learning, and feel this is worthy of reporting in its own right.

From this perspective, our online decoding approach allows the monitoring of brain dynamics and has the power to detect, in a reliable manner, representations distributed across patterns of neural activity that are at the same time specific and unconscious. Task conditions are not defined externally through images or other media, but internally - and we can directly test the ability of human participants to learn to read their own mental states.

As highlighted by the reviewer, the adoption of such a design does suffer from one main disadvantage: namely, the lack of an explicit experimental manipulation of the relevant decision variables, thus precluding a strong causal interpretation of metacognition in learning from rewards.

An experiment in which task accuracy and confidence are independently manipulated would be very interesting, and could follow nicely on this and previous work on conscious perception. But these two approaches test different questions - ours tests the ability of the brain to learn distributed, unconscious mental representations to garner rewards, while the other would test the effect of lower/higher confidence in learning rewarding choices. This would be a new study with a different goal from the current work.

We now clarify that the main question of our work was not to test the directional link between metacognition and RL. The causal relationship between metacognition and RL is discussed only in the context of it being a possible mechanism, given the surprising and positive finding that subjects were able to learn the task.

Is this causal mechanism of metacognition for RL plausible? As a secondary finding, we think our results support its plausibility. While the correlational nature cannot be avoided here, our results also suggest the direction metacognition → RL for the following reasons: (1) initial metacognitive ability is predictive of future baseline RL task performance, (2) in sessions 1-2, receiving a reward or not has no effect on the next trial's confidence judgement.

Given these considerations, we have:

- 1) Clarified what is the goal of using the decoding approach in this context (introduction, e.g. lines 64 - 70, 87 - 89)*
- 2) Clarified what are the advantages and disadvantages of the task design adopted in this study (discussion, lines 502 - 509)*
- 3) Softened the overall interpretation of the results and the strength of the claims with regards to confidence/metacognition, putting them in a more speculative light. These changes are reflected in the title, introduction, results and discussion.*

2/ The Reviewer suggests that the problem faced by participants in the task is not high-dimensional, and identifies several factors that could make learning much easier.

We agree with the reviewer that the problem is unlikely to be fully unconstrained and that this argument may be toned down. However, we should stress the following aspects of the task to make the case that it is, indeed, not a trivial problem, but actually an astronomically large problem.

Even though participants should be clearly clued towards the fact that task contingencies are related to motion direction, their belief is that this depends on the motion direction they see - and not the motion direction inferred in real-time from their patterns of brain activity. Since these patterns are effectively measured during the pre-stimulus period, the possibility that imagination, or illusory perception could have indexed the decoder output is minimal. The patterns of neural activity thus measured are less constrained than those occurring in response to a visual stimulus (see supplementary figure S11). Moreover, the decoders have an average cross-validated accuracy of ~70-75%, which means that sometimes similar patterns of activity may be categorized as two different states. Finally, the decoders are based on a sparse number of voxels within a given brain region. To the brain, neither the location of activity nor the voxels themselves are known. The high-dimensionality of the problem stems precisely from this: the brain has to find the decoder boundary from patterns of activity of neurons, although this boundary is determined through voxels, and only through some of them. The brain has no possibility to know which voxels are directly

relevant - especially since in some cases these voxels are in the visual cortex and in other cases in the prefrontal cortex. Suppose that 10,000 voxels are somehow activated in these RL tasks. The machine learning decoder selected about 100 voxels for each participant. The combinatorial number to select 100 out of 10,000 is astronomical. Even with several constraints such as redundancy in voxels' activities or particular patterns related to motion, participants still have to efficiently find the correct states within a huge space.

Binding these lines of thought together, it is possible to appreciate the inherent difficulty of the task in terms of dimensionality, that goes beyond a (trivial) binary problem of motion directions. Ultimately, if the problem was simple, participants gambling performance would have been higher.

We have added a supplementary figure (supplementary figure S11) showing that multivoxel patterns during early visual stimulus are more constrained than those from the pre-stimulus period as they account for more of the signal's variance. We have also amended the introduction and discussion sections to describe the above points in clearer terms, particularly to highlight that while the problem is complex and high-dimensional, it is also not fully unconstrained, acknowledging possible external solutions that could make the problem easier to learn (introduction, line 72; discussion, lines 522 - 567).

3/ The computational modelling is a good validation of the role of RL in describing gambling performance, as well as for the relationship between confidence and RL. However, the modelling does not provide any additional support for the causal role of confidence in RL.

We agree that the general presentation of the relation metacognition → RL could be toned down, as it is also secondary. However, behavioural data indicate that metacognitive ability is predictive of future RL performance (figure 4A), and that rewards alone cannot explain fluctuations in confidence judgements (supplementary figure S5D), hinting towards a potential causal role of metacognition in learning. We acknowledge that the modelling part does not, in itself, strengthen the case for the causal relevance of metacognition for learning, rightfully raised by the Reviewer. What the modelling does is to validate the intuition that participants learn to use the latent spaces and provides additional support for the trial-by-trial relationship between confidence and subsequent prediction errors. The modelling also shows that the reverse direction of prediction error affecting the next trial's confidence is weaker and noisier. Although the evidence is indirect, considering the above results, the direction metacognition → RL remains the most likely interpretation.

Overall, we have followed the reviewer's suggestion and revised accordingly, at several levels, the manuscript. We have changed the title, as well as text within the introduction (i.e., lines 72 - 89), results (i.e., lines 276 - 279, 285 - 287) and discussion (i.e., lines 470 - 501) to keep the interpretation on metacognitive functions more nuanced and speculative.

4/ The neuroimaging results are qualitative, and it is difficult to see how they can provide specific insight into the claims of the paper. The 3rd session is different from the first two, making comparisons confusing. Furthermore, resting state connectivity changes due to learning over the three sessions cannot be well separated from changes due to time.

We acknowledge the limitation of the original neuroimaging results previously reported in figure 6. Of note, in session 3 the latent states were revealed to participants, but the coherence was very low for the first half of the session (we have added this information in the main text as originally it was only in supplementary). Participants did not realize that motion direction was the relevant state until late in the session, when coherence was very high at around 50%. This aspect indicates the inherent difficulty of the RL task even in session 3.

To avoid basing the study's neural interpretations solely on qualitative neuroimaging results, we have moved the analysis that combines decoding of confidence in DLPFC and |RPE| in the basal ganglia from supplementary to the main text (new main figure 6). This analysis presents in quantitative terms the increasingly strong functional relationship between confidence and RL at the (neural) multivoxel level.

Furthermore, acknowledging that resting-state connectivity changes over the three scans may partly reflect correlated but unrelated factors, and considering that in session 2 subjects showed evidence of learning while the task remained high-dimensional and subconscious, we now restrict the resting-state analysis to the two scans immediately before and after session 2 (supplementary figure S9). Results provide more support to the main claim of a link between metacognition and RL during learning: increased functional connectivity was found between the seed region in the basal ganglia and right DLPFC and inferior parietal lobule. Both regions have been previously linked with confidence judgements and metacognition (Rounis et al. 2010, Qiu et al. 2018) and validate the main neural result reported in figure 6B-C.

5/ The sample size is small and makes the presence or absence of group differences difficult to interpret.

N=18 is indeed a relatively low sample size, if considered as single samples. However, as in vision psychophysics studies, sometimes intensive studies of a few individuals can be beneficial. Ultimately, what may matter could be the total amount of data collected on the phenomenon. In the present study N=18 subjects participated in 3 scanning sessions, totalling 54 hours of fMRI data collection.

Although such repeated measures data formats may not be exactly common, the aggregate N=27 per group is in line with recent work that investigated group differences (Iwata et al. 2018, Levitt et al. 2020, Losin et al. 2019).

Also, in fact, group differences were detected in our study, and we apologize that this information was previously lost within a single line in the main text. In the unsigned

|RPE| - confidence analysis, there was a relatively strong interaction between factors 'confidence' and 'group' on the dependent variable |RPE| ($p=0.004$). We have now added a new panel to figure 5 to provide a more in-depth look at this effect of confidence on |RPE| based on a subject's decoder (VC or PFC). This new subdivision of the data revealed a much larger effect of confidence in the PFC group, which we now discuss in the main text (results, lines 344 - 350; discussion, lines 482 - 484).

Considering the lack of significant correlation between task performance and discrimination accuracy, we now explicitly refer to it as a trend in the main text, (results, lines 137-142; discussion, lines 517-521). The fact that the effect was a trend and not significant should not be taken as an absence of effect but rather as a small effect - which, per se, does not invalidate the main claim.

Reviewer #3 (Remarks to the Author):

1. The Reviewer pointed out that claims about confidence facilitating RL state search should be more speculative, as the data is observational. The Reviewer added that successful search could be, in fact, driving confidence expression.

We are sorry for the confusion, but the primary objective of the present study was to test if humans can learn a task in which the information that determines the RL states is (a) high dimensional and (b) unconscious. The goal was to assess whether such learning is at all possible under such extreme circumstances, while the general relationship between metacognition and RL was secondary. The possibility of metacognition being causally involved was only brought up because the primary objective of the study was met - surprisingly we observed such learning, and feel this is worthy of reporting in its own right.

Yet, although correlational, three lines of evidence, detailed hereafter, indicate that the direction metacognition → RL remains the most likely explanation. (1) initial metacognitive ability computed with prior task data can predict baseline success in the RL task (figure 4A). This data cannot be explained by the opposite direction link from search success to metacognition. (2) Additionally, if it were purely that search success influences confidence, whether a reward was delivered or not at trial t should affect the subsequent confidence judgment at $t+1$. This was not the case in sessions 1-2 (supplementary figure S5D). (3) Of course, we cannot disprove some common intelligence factor which affects both metacognition and RL capability in an equal manner. But this is again not compatible with PFC and BG information connection (figure 6B-C), which is a trial-wise analysis with much higher temporal resolution. Although we do not have direct evidence, considering the above results the direction metacognition → RL remains the most likely interpretation.

Nevertheless, particularly because the evidence is indirect, we have modified the manuscript accordingly to make this specific interpretation more nuanced and speculative and open for alternative explanations: the title, introduction (i.e., lines 78 - 89), results (lines 276 - 279, 285 - 287), and discussion (i.e., lines 464 - 493). In the discussion we now also comment on the alternative model, in which confidence is modulated by successful search (lines 470 - 474).

2. Several of the main results are significant at a permissive threshold ($p \sim 0.01$), which poses issues of multiple comparisons. The same concern applies to those neuroimaging results reported at $p < 0.005$ uncorrected. The Reviewer suggests to run the same experiment in a replication sample.

We agree that a replication sample would be the most direct way to quell any doubt about the exploratory nature of the study and statistical validity of the results. Nevertheless, given the very large effort involved (particularly in terms of monetary costs), we have instead considered the following measures to answer these concerns.

- 1) *As kindly suggested by the Reviewer in comment (3.), to increase robustness of the behavioural data analyses, all tests against chance are now implemented with full linear models (intercept + random effects) where the intercept is the difference from chance and subjects are modelled as random effects. This approach strengthens the statistical validity of the results as individual subjects are treated as random effects.*
- 2) *Related to comment #6, we have changed the correlation between meta-d' and sum of rewards in sessions 1-2 to meta-d' and baseline task performance in sessions 1-2. This measure strengthens the statistical validity of the metacognition - RL link since it goes beyond a mere correlation and indicates that metacognitive ability is predictive of future baseline RL performance.*
- 3) *Neuroimaging results are now reported at $p(\text{FPR}) < 0.001$ and cluster size $k > 30$ (RPE regression); $p(\text{unc.}) < 0.001$ and cluster size $p(\text{FDR}) < 0.05$ (resting state). Furthermore, with regard to resting state data, we have restricted the analysis to the two scans performed before and after session 2. The rationale is that session #2 showed the largest learning effect in the unconscious condition, and by focusing on two resting-state scans only, the results are less likely to be due to other, unrelated, factors, thus leading to increased power. Given their descriptive nature, we have moved these neuroimaging results to supplementary while referencing and discussing them in the main text (e.g., lines 369 - 398).*
- 4) *Finally, we have moved the previous supplementary figure 8 to the main text (current figure 6), as it supports, in statistically much stronger terms, the association between metacognition/confidence and RPE (at the neural level).*

3. The Reviewer recommends to correct instances related to the issue of “difference in significance does not imply significant difference”. The Reviewer further recommends to use full linear models when testing against chance level.

We have modified the text accordingly. Following the reviewer’s kind suggestion, we have changed from simple statistical tests to full linear models (intercept as difference from chance, and subjects as random effects) in all cases requiring testing against chance level.

4. The Reviewer indicated that it is not clear what “perceptual discrimination” in session 1 and 2 means, and recommended to provide clearer definitions of the dependent variables.

By perceptual discrimination we meant that subjects were instructed to report the direction they perceived. Although stimuli were purely random, subjects were instructed that stimuli were very noisy and dim and that it would be difficult to tell the direction, so as to keep them engaged in the task. This definition is now provided in the introduction (lines 58 - 60). We hope the current version of the manuscript shows clearer definitions of the dependent variables.

5. The trend between accuracy in the two tasks seems problematic, particularly because the difference between the two tasks is not large. Discrimination accuracy could be entered as a control for $p(\text{opt action})$.

As the reviewer points out, if one considers discrimination accuracy, action selection may not show any significant conditioning. We have run this control analysis, and found that on session 2, even after entering discrimination accuracy as covariate, $p(\text{opt action})$ remains significantly different from chance ($p=0.005$). On session 3, the significant effect on $p(\text{opt action})$ is lost, because discrimination accuracy itself is very high, highlighting the issue with this approach: even if participants have fully learned the gambling task, if the discrimination is very high the former will still not show any sign of conditioning. Of note, the difference between the two tasks, if including all sessions, is large: the task difference is significant at $p=0.0003$.

Yet, a correlation between the accuracy in the two tasks does not invalidate the conditioning or the learning in the gambling task. Indeed, a correct discrimination is also useful for the subsequent gambling action selection. As such, we can imagine an agent that is truly unconscious, yet discriminates above chance and also chooses the most rewarding option. In line with this series of thoughts, and as has been similarly documented in the perceptual learning domain (Seitz et al. 2009), the effect of rewards in one task/dimension could have leaked to the other (unrewarded) task/dimension. This point is now discussed in the main text (lines 517 - 521).

6. It is unclear why the metric ‘sum of rewards’ rather than ‘ $p(\text{opt action})$ ’ was used to show the link between learning and metacognition. It is also unclear whether the prediction holds for both the PFC and VC groups.

The metric ‘sum of rewards’ was used to provide one additional line of result toward the claim that metacognition correlates with learning. We acknowledge that this may actually have the opposite effect and confuse the reader. As such, we now report meta- d' as predictive of the baseline $p(\text{opt action})$ on sessions 1-2 ($N=18$, $r=.56$, $p=.017$).

Taking subjects separately according to their decoder location, the predictive power of metacognitive ability held only for subjects in the PFC group ($N=9$, $r=.72$, $p=0.03$), but not so for subjects in the VC group ($N=9$, $r=.51$, $p=.16$). The two coefficients were, however, not significantly different. While these results are very interesting and hint at a greater effect of PFC, in this case the result prevents any immediate strong conclusion. This is explicitly reported in the main text (lines 304 - 311) and discussed (lines 482 - 484).

Incidentally, throughout all analyses, subjects were always counted once, as there was no case in which a subject was counted twice within a plot or analysis, unless explicitly accounted for in a linear mixed effect model. The total $N=18$ subjects included one group of subjects with decoder in VC ($N=9$), and one group with decoder in PFC ($N=9$), but these were different subjects.

7. The authors regress RPE against brain activity (model-based fMRI), but see Lebreton et al. (Nature Human Behavior; 2019) for issues related to this technique. In

particular the authors should specify whether the RPE regressors have been z-scored, across subjects and/or session, justify their approach and specify whether the results are affected by their analytical choices.

We thank the reviewer for the very useful comment. We have clarified the approach we used in the main analysis (lines 376/377, 384 - 386), and we have also added in supplementary material a confirmation analysis where RPE regressors were z-scored across subjects and sessions (supplementary figures S7, S8).

Minors

The authors equate reward contingencies to RL states. I see why this is the case in their design, but different states may have the same reward contingencies and vice versa. Please consider avoiding this terminology confusion.

This has been clarified throughout the manuscript.

Oddly, the result session starts with a statement about the interpretation of the results, before providing any relevant quantitative and statistical evidence. Please consider replacing this sentence with more factual information.

Good point, this has been corrected.

“implicit metacognition” is mentioned in the abstract, but not well defined. The definition given in the abstract is not clear for a scientist not expert in meta-cognition.

Good point, the definition has been changed to be accessible for a broader readership.

In the abstract there is mention of possible applications of these findings in artificial intelligence. This is a pretty wild speculation (the models proposed are very simple, computers do not have brains and the role of meta-cognition in AI is very much debated), so please remove it from the abstract.

The last sentence of the abstract has been changed.

The term “brain states” should be clarified at its first occurrence (in the instruction). Different neuroscientists will have very different interpretations of what a brain state is.

Good point, this has been clarified.

The term “discrimination confidence” should be clarified at its first occurrence (in the instruction). It is unclear for a scientist outside the meta-cognition field.

Good point, this has been clarified.

REVIEWER COMMENTS

Reviewer #1 (Remarks to the Author):

Review of "Learning to use latent brain representations with confidence"

In this revised manuscript the authors have toned down claims about the causal role of meta-cognition in the ability to learn latent state representations from brain states. The manuscript is considerably improved in the light of the revisions made.

A few outstanding issues are noted:

(1) The division of confidence ratings into different high vs low categories for session 1 and 2 (1 = low, 2-4 = high) vs session 3 (1 to 3 = low, 4 = high) shown in Fig 5B seems arbitrary and difficult to justify other than by the fact that dividing the data differently in this way shows a significant differential effect between PFC and VC. The authors should either use a consistent criterion across the 3 sessions if they wish to make claims about common effects across sessions, or they need to justify why they have elected to use different criteria in the different sessions. Session 3 is a different task altogether from session 1 and 2, so there may be a justification for doing so, but if they want to claim common mechanisms, using the same analysis criteria across the two different tasks would seem prudent.

(2) The neural substrates results presented on page 14 refer to supplementary figure s7 in which it is claimed that the RPE signal describes a global search in sessions 1 and 2 which eventually converges to the basal ganglia by session 3. However, this convergence may not be anything to do with a convergence of a global search – instead session 3 is a different task to sessions 1 and 2, and so any difference in RPE correlates between sessions 1 and 2 vs session 3 could simply reflect the difference in the task properties, and be nothing to do with a convergence in global search. If they find evidence for such convergence WITHIN sessions 1 and 2 without considering session 3, then this claim could be made more strongly perhaps. As is, it should not be made without severe qualification, because it is impossible to tell why session 3 is different to sessions 1 and 2, the most parsimonious explanation being it is due to the difference in the task.

Reviewer #2 (Remarks to the Author):

I continue to feel that this study is an interesting one that will command considerable interest. In brief, what the Cortese and colleagues have done is very unusual. They have shown that people can learn to take one of two different actions conditional on the existence of one of two states. Such conditional learning is well known but what makes this report exciting is the nature of the states being considered – they are multivariate patterns of activity distributed across voxels in either prefrontal cortex or visual cortex. Just the observation that human subjects are able to do this will be of interest to many readers. In their response to reviewer 1 point 2 the authors have summarized very clearly why this is an unusually interesting result to have obtained; it is remarkable that the participants were able to identify that these internal brain states defined the conditions that determined which action to take.

Second, the authors have shown that there is a relationship between the learning of this process and a measure of subjective confidence that the participants provide. Some of the other reviewers have focussed on the direction of cause and effect underlying this correlation. This is indeed an interesting question but it is secondary to the fact that the correlation exists. In response to the reviewers, the authors have toned down their description of this aspect of the results so that the correlational

findings are presented clearly. Nevertheless, there are interesting reasons for thinking that the authors maybe correct to speculate that better metacognition leads to better learning. Even if the authors' interpretation of the direction of cause and effect proves ultimately to be incorrect the observation of the correlation will remain interesting and is likely to drive future investigation. As it stands it is reported in a measured manner.

Throughout the report the authors have illustrated the neural data on which learning is based. Reviewer 1 has argued that the neuroimaging data are of little value but, of course, the neural data are critical here. They serve in the place of stimuli that might be learned about. The neuroimaging data are then used in quite a different manner in figure 6 which summarizes a very different set of neural activity patterns and their evolution during learning. These are not the patterns that the participants were learning about but rather they are patterns related to the learning process itself: patterns related subjective confidence in decisions and patterns related to the prediction errors that drive learning.

Finally, it might be worth noting that the authors have scanned a reasonably sized group of human participants but they have scanned each person three times so that they are working with a data set that is much larger than the average behavioural of neuroimaging data set.

Reviewer #3 (Remarks to the Author):

I am not quite satisfied by the authors' response concerning the problem of sample size (issue 2). The fact that scanning is time consuming and expensive is not recevable as a scientific argument. They start with 22 subjects and exclude 4 subjects (3 for technical reasons, 1 because the decoder was not working). This means that they excluded 18% of the initial subjects. The final sample size of 18 is far below the currently accepted sample size. A recent meta-analysis (Lebreton et al. 2019) indicates that the median sample size for model-based fMRI studies in the past years is 27, meaning that their sample size represents the 66% of the average accepted sample size. If they do not want to run a replication experiment, they cold at least increase the sample size to match the median one. I also ask them to perform retrospective power calculation to determine what is the power of detecting the reported effects with their sample size.

POINT-BY-POINT RESPONSE TO THE REVIEWERS:

We thank the Reviewers for their additional comments. We find these to be helpful to further ameliorate the quality of the manuscript. Here, we answer to the points that were raised.

Reviewer #1 (Remarks to the Author):

In this revised manuscript the authors have toned down claims about the causal role of meta-cognition in the ability to learn latent state representations from brain states. The manuscript is considerably improved in the light of the revisions made. A few outstanding issues are noted:

- (1) The Reviewer pointed out that the different division of confidence ratings into 'high' and 'low' categories for sessions 1-2 vs session 3 was not justified in the manuscript. The reviewer recommends to either justify the difference or use the same criterion across all sessions.**

We thank the reviewer for noticing the issue, as we had overlooked to provide a justification for the trial splitting into 'low' and 'high' confidence categories. The main reason behind this division was due to the observation that in sessions 1 and 2 participants tended to use more lower confidence ratings, while in session 3 participants tended to use high confidence ratings (i.e., majority of 4s). A reasoned way to implement the above observation is to split trials according to the median confidence in each session. This allows for possible session-specific differences in criteria in how participants rated their confidence. We now report in the main text as well as in the legend of figure 5b that the labelling is based on the median confidence level in each session, labelling as 'low' the trials with confidence below the median, and as 'high', those with confidence equal or above (since in session 3 the median is 4). We now directly plot the bootstrapped |RPE| difference between high and low confidence trials, for each group (PFC vs VC) and session. The results remain qualitatively and quantitatively unchanged, confirming the differential effect between PFC and VC.

- (2) The differences in neural correlates between sessions 1, 2, and 3 may not be due to a convergence of a global search but rather, more parsimoniously, to simple differences in task properties.**

We agree that the interpretation of this result is not univocal, the simplest explanation being that differences in neural correlates between sessions 1-2 and 3 are due to task differences. As such, to better balance the manuscript and data interpretation, we have amended the main text. We now first introduce the most parsimonious explanation (i.e., task differences), and only after speculate about the possibility this may (also) reflect a convergence of a global search for task-states.

Reviewer #2 (Remarks to the Author):

I continue to feel that this study is an interesting one that will command considerable interest. In brief, what the Cortese and colleagues have done is very unusual. They have shown that people can learn to take one of two different actions conditional on the existence of one of two states. Such conditional learning is well known but what makes this report exciting is the nature of the states being considered – they are multivariate patterns of activity distributed across voxels in either the prefrontal cortex or visual cortex. Just the observation that human subjects are able to do this will be of interest to many readers. In their response to reviewer 1 point 2 the authors have summarized very clearly why this is an unusually interesting result to have obtained; it is remarkable that the participants were able to identify that these internal brain states defined the conditions that determined which action to take.

Second, the authors have shown that there is a relationship between the learning of this process and a measure of subjective confidence that the participants provide. Some of the other reviewers have focussed on the direction of cause and effect underlying this correlation. This is indeed an interesting question but it is secondary to the fact that the correlation exists. In response to the reviewers, the authors have toned down their description of this aspect of the results so that the correlational findings are presented clearly. Nevertheless, there are interesting reasons for thinking that the authors may be correct to speculate that better metacognition leads to better learning. Even if the authors' interpretation of the direction of cause and effect proves ultimately to be incorrect the observation of the correlation will remain interesting and is likely to drive future investigation. As it stands it is reported in a measured manner.

Throughout the report the authors have illustrated the neural data on which learning is based. Reviewer 1 has argued that the neuroimaging data are of little value but, of course, the neural data are critical here. They serve in the place of stimuli that might be learned about. The neuroimaging data are then used in quite a different manner in figure 6 which summarizes a very different set of neural activity patterns and their evolution during learning. These are not the patterns that the participants were learning about but rather they are patterns related to the learning process itself: patterns related subjective confidence in decisions and patterns related to the prediction errors that drive learning.

Finally, it might be worth noting that the authors have scanned a reasonably sized group of human participants but they have scanned each person three times so that they are working with a data set that is much larger than the average behavioural or neuroimaging data set.

We thank the Reviewer for their clear summary in support of this work. We are happy that our excitement for this work is shared, and hope that many readers will feel the same way.

Reviewer #3 (Remarks to the Author):

I am not quite satisfied by the authors' response concerning the problem of sample size (issue 2). The fact that scanning is time consuming and expensive is not recevable as a scientific argument. They start with 22 subjects and exclude 4 subjects (3 for technical reasons, 1 because the decoder was not working). This means that they excluded 18% of the initial subjects. The final sample size of 18 is far below the currently accepted sample size. A recent meta-analysis (Lebreton et al. 2019) indicates that the median sample size for model-based fMRI studies in the past years is 27, meaning that their sample size represents the 66% of the average accepted sample size. If they do not want to run a replication experiment, they could at least increase the sample size to match the median one. I also ask them to perform retrospective power calculation to determine what is the power of detecting the reported effects with their sample size.

We appreciate the Reviewer's concern on this important point. It is true that a sample size of 18, under normal conditions of one scanning session each, would be rather small. However, considering that 18 subjects were scanned 3 times each, the data collected is in fact larger than the average human neuroimaging study (54 scanning sessions). Furthermore, in psychophysics studies it is common to work with smaller groups of subjects, balanced by a higher number of sessions/trials. We refer the Reviewer to our previous replies (e.g., also to Reviewer 1) where we provided more detailed answers. The fact that fMRI scanning is time consuming and expensive is indeed, not a scientific argument, and was only meant as a pragmatic comment given that the actual sample size of $18 \times 3 = 54$ is considerably large.